TOPICAL REVIEW

# Advancing respiratory–cardiovascular physiology with the working heart–brainstem preparation over 25 years

Julian F. R. Paton[1] , Benedito H. Machado[2], Davi J. A. Moraes[2], Daniel B. Zoccal[3], Ana P. Abdala[4], Jeffrey C. Smith[5], Vagner R. Antunes[6], David Murphy[7], Mathias Dutschmann[8], Rishi R. Dhingra[8], Robin McAllen[8], Anthony E. Pickering[4], Richard J. A. Wilson[9], Trevor A. Day[9,10], Nicole O. Barioni[9], Andrew M. Allen[11], Clément Menuet[12], Joseph Donnelly[13], Igor Felippe[1] and Walter M. St-John[14]

[1]*Manaaki Manawa – The Centre for Heart Research, Faculty of Medical & Health Science, University of Auckland, Grafton, Auckland, New Zealand*
[2]*Department of Physiology, School of Medicine of Ribeirão Preto, University of São Paulo, Ribeirão Preto, São Paulo, Brazil*
[3]*Department of Physiology and Pathology, School of Dentistry of Araraquara, São Paulo State University, Araraquara, São Paulo, Brazil*
[4]*School of Physiology, Pharmacology and Neuroscience, Faculty of Biomedical Sciences, University of Bristol, Bristol, UK*
[5]*Cellular and Systems Neurobiology Section, National Institute of Neurological Disorders and Stroke, National Institutes of Health, Bethesda, MD, USA*
[6]*Department of Physiology and Biophysics, Institute of Biomedical Sciences, University of São Paulo, São Paulo, Brazil*
[7]*Molecular Neuroendocrinology Research Group, Bristol Medical School: Translational Health Sciences, University of Bristol, Bristol, UK*
[8]*Florey institute of Neuroscience and Mental Health, University of Melbourne, Parkville, Victoria, Australia*
[9]*Department of Physiology and Pharmacology, Hotchkiss Brain Institute and Alberta Children's Hospital Research Institute, Cumming School of Medicine, University of Calgary, Calgary, Alberta, Canada*
[10]*Department of Biology, Faculty of Science and Technology, Mount Royal University, Calgary, Alberta, Canada*
[11]*Department of Anatomy & Physiology, University of Melbourne, Victoria, Australia*
[12]*Institut de Neurobiologie de la Méditerranée, INMED UMR1249, INSERM, Aix-Marseille Université, Marseille, France*
[13]*Department of Medicine, Faculty of Medical and Health Sciences, University of Auckland, Auckland, New Zealand*
[14]*Emeritus Professor, Department of Physiology and Neurobiology, Geisel School of Medicine at Dartmouth, Dartmouth, NH, USA*

Edited by: Ian Forsythe & Frank Powell

The peer review history is available in the Supporting Information section of this article (https://doi.org/10.1113/JP281953#support-information-section).

J. F. R. Paton and B. H. Machado are joint first authors.

**Abstract** Twenty-five years ago, a new physiological preparation called the working heart–brainstem preparation (WHBP) was introduced with the claim it would provide a new platform allowing studies not possible before in cardiovascular, neuroendocrine, autonomic and respiratory research. Herein, we review some of the progress made with the WHBP, some advantages and disadvantages along with potential future applications, and provide photographs and technical drawings of all the customised equipment used for the preparation. Using mice or rats, the WHBP is an *in situ* experimental model that is perfused via an extracorporeal circuit benefitting from unprecedented surgical access, mechanical stability of the brain for whole cell recording and an uncompromised use of pharmacological agents akin to *in vitro* approaches. The preparation has revealed novel mechanistic insights into, for example, the generation of distinct respiratory rhythms, the neurogenesis of sympathetic activity, coupling between respiration and the heart and circulation, hypothalamic and spinal control mechanisms, and peripheral and central chemoreceptor mechanisms. Insights have been gleaned into diseases such as hypertension, heart failure and sleep apnoea. Findings from the *in situ* preparation have been ratified in conscious *in vivo* animals and when tested have translated to humans. We conclude by discussing potential future applications of the WHBP including two-photon imaging of peripheral and central nervous systems and adoption of pharmacogenetic tools that will improve our understanding of physiological mechanisms and reveal novel mechanisms that may guide new treatment strategies for cardiorespiratory diseases.

(Received 20 October 2021; accepted after revision 4 March 2022; first published online 16 March 2022)

**Corresponding author** J. F. R. Paton: Manaaki Manawa – The Centre for Heart Research, Department of Physiology, Faculty of Medical & Health Sciences, University of Auckland, Park Road, Grafton, Auckland, 1142, New Zealand. Email: j.paton@auckland.ac.nz.

**Abstract figure legend** The working heart–brainstem preparation (WHBP) was first published in 1996 and since that time has revealed novel mechanisms in the generation of breathing rhythms and patterns, autonomic control of the heart and circulation, and the physiological relevance of the coupling between respiratory and cardiovascular systems. This review brings together an international cohort of authors who have adopted the WHBP, to highlight some of the major advances and discoveries made using this *in situ* arterially perfused preparation.

## Introduction

In 1995, a new preparation for integrative physiological research was presented to The Physiological Society at a meeting held at University College Cork between 19 and 22 September (Paton, 1995). This was called the working heart–brainstem preparation (WHBP), and was demonstrated 'live' at The Physiological Society meeting at the University of Bristol in 1996; a full account of the methods was published subsequently (Paton, 1996a; Fig. 1). This review highlights the subsequent contributions that the WHBP has made to understanding the neural mechanisms regulating the respiratory and cardiovascular systems and their coupling in health and disease.

We initiate our discussion with the original reason for developing an *in situ* preparation, the value of the preparation as a teaching approach to integrative physiology, followed by exemplars from a range of investigators

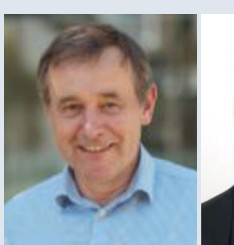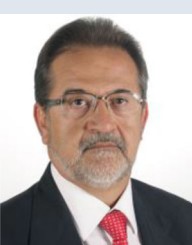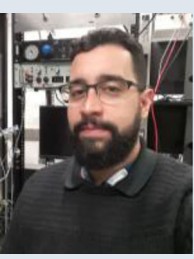

**Julian F. R. Paton**, PhD, FRSNZ, is a Professor of Translational Physiology at the University of Auckland, Aotearoa, New Zealand. He is Director of Manaaki Manawa – the Centre for Heart Research in Auckland. He bases his research on animal models of disease to inform new potential ways to control hypertension and alleviate heart failure and sleep apnoea. **Benedito H. Machado**, PhD, is Professor of Physiology at the School of Medicine of Ribeirão Preto, University of São Paulo, Brazil, and the main focus of his current projects is related to changes in the synaptic transmission of autonomic and respiratory neurons integral to the chemoreflex pathways at the brainstem of rodents in response to physiological challenges. **Igor Felippe** is a PhD candidate supervised by Julian Paton at the University of Auckland. Originally from Vitória-ES in Brazil, his research interest encompasses the understanding of carotid body physiology and its involvement in the development of hypertension and the toxicological effects of organophosphate pesticides.

around the globe. The exemplars illustrate a broad range of research questions, diverse and ingenious approaches that demonstrate the versatility of the preparation and the substantial advance in new knowledge, some of which has contributed to first-in-human studies.

## Why was the WHBP developed?

At the time there were no *in vitro* preparations to study the cardiovascular system, yet there were for the respiratory system (Smith et al., 1991; Suzue, 1984). The neural genesis, regulation and co-ordination of the yin and yang autonomic limbs has been of longstanding academic interest (Paton et al., 2005) and of fundamental importance in the quest to identify novel targets for therapeutic intervention. The entire essence of the WHBP was to establish a preparation that would maintain advantages of contemporary acute *in vivo* and *in vitro* approaches but would circumvent their limitations. It is important to recognise that before the WHBP there were a number of arterially perfused preparations (Hayashi

et al., 1991; Llinas & Muhlethaler, 1988; Morin-Surun & Denavit-Saubie, 1989; Muhlethaler et al., 1993; Richerson & Getting, 1987), but they were not deemed suitable at the time because they contained the brain only and/or were without sensory receptors, motor nerves and their target organs, and therefore did not allow the study of neural mechanisms regulating the cardiovascular system. Some of these previous preparations used an oxygen carrier, perfluorotributylamine (FC-43), which was expensive and unavailable when the WHBP was developed (Hayashi et al., 1991). The use of these previous preparations was limited, as observed from the one or two publications that resulted from the inventors only. The idea for the WHBP was that it should be easily transferable to any interested investigator, and this has proved to be the case as we know of at least 25 laboratories around the world that are using the preparation and >200 original research papers published.

Table 1 summarises the technical limitations of contemporary preparations that drove the design of the WHBP. The main drivers were to stabilise the brainstem from cardiac pulsing (arterial and venous) and

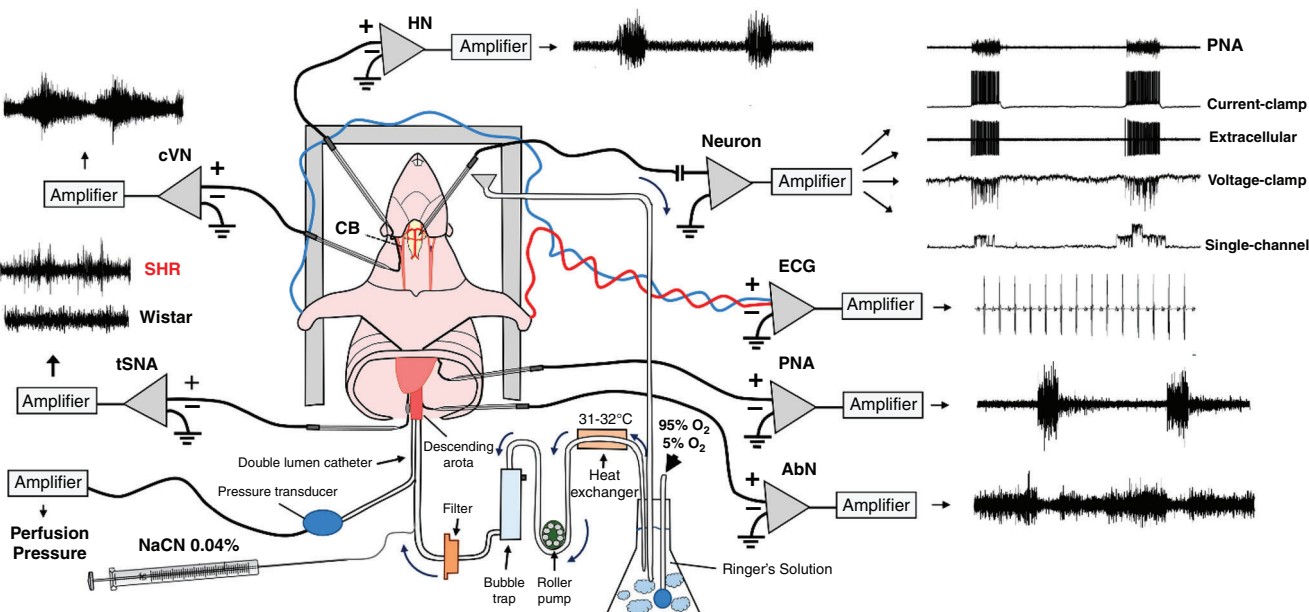

**Figure 1. The WHBP**
Schematic representation of the working heart–brainstem preparation, illustrating multiple respiratory and sympathetic nerves that can be recorded simultaneously using bipolar suction electrodes. Note the preparation has an open circulation allowing perfusate to be re-circulated using a roller pump. Perfusate (Ringer plus oncotic agent) is carbogenated, warmed to 31°C, filtered (25 mm pore nylon mesh) and bubbles removed in Perspex traps prior to delivery via a double lumen catheter placed into the descending aorta for retrograde perfusion; perfusion pressure is measure at the tip of this catheter from the second lumen. See Supporting information for drawings of all apparatus for the WHBP. This catheter has a side port for administration of drugs; shown here is potassium cyanide to activate peripheral chemoreceptors. A neuromuscular blocking agent is added to the perfusate to prevent respiratory-related movement. Using a ventral exposure, extracellular and/or patch pipettes can be placed into the medulla oblongata for single neuron recording. Notable landmarks such as hypoglossal rootlets, trapezoid body and arteries allow accurate positioning of electrodes into the brain. It is possible to remove cytoplasm for single cell qRT-PCR. Abbreviations: AbN, abdominal nerve; cVN, cervical vagus nerve; KCN, potassium cyanide; PN, phrenic nerve; tSN, thoracic sympathetic nerve.

**Table 1. Ideas that drove the invention of the working heart–brainstem preparation**

| Limitation | Solution provided by WHBP |
| --- | --- |
| Depressant effect of anaesthesia *in vivo* | Decorticate or decerebrate, so insentient |
| Mechanical stability of brain for intracellular sharp microelectrode and whole cell patch recording | Constant low perfusion pressure, open chest, open circulation, reduced venous pulsing |
| Matching oxygen demand without an expensive oxygen carrier molecule | Standard Ringer solution, low viscosity, high flow, reduced temperature |
| Oedema when not using blood | Oncotic agent (not albumin, which froths) |
| Manageable perfusate volume | Open circulation and re-circulation of perfusate |
| Surgical access for fine nerves and those in the chest | Unlimited as severance of small vessels not an issue |
| Hygiene | Remove fur/skin; clean and dry extracorporeal circuit overnight |
| Physiological viability readout of brainstem | Record respiratory motor pattern (phrenic) or better, central vagus also for post-inspiration |
| Brainstem transections and fatal haemorrhage | Absence of blood and unlimited blood volume allow transections of the neuraxis |
| Time taken to set up an *in vivo* preparation | 20–30 min set-up time for the WHBP |
| Long washout time of drugs prevents reversibility of effect | Refreshing the perfusate allows drug washout instantly |
| Constant blood gas sampling to ensure physiological $P_{aCO_2}$, pH and $HCO_3^-$ | Perfusate allows clamping of gas partial pressures, pH and $HCO_3^-$ |
| Only drugs that do not collapse or poison the cardiovascular or respiratory systems can be used | Any drug including poisons and blockers of neurotransmission as preparation viability is supported by extracorporeal perfusion |
| Ligature of the coronary arteries to evaluate the impact of ischaemia and activation of cardiac afferents may affect the haemodynamics | Preparation viability is supported by extracorporeal perfusion despite cardiac dysfunction |

respiratory-related movement so that long term intracellular recordings could be made using sharp microelectrodes from cardiovascular and respiratory neurones (Paton, 1996b, 1997) and subsequently patch clamp recordings (e.g. Dutschmann & Paton, 2003; Paton & St-John, 2005; Paton et al., 2001). To identify physiologically sub-populations of cardiovascular/sympathetic neurones, it was essential that sensory receptors could be stimulated physiologically (e.g. baroreceptors, cardiac receptors, skeletal muscle; Boscan & Paton, 2001; Paton, 1998; Potts et al., 2000) and/or their activity correlated with a motor nerve output and/or end organ response (Koganezawa & Paton, 2014). Additional examples are highlighted later in the review of how the preparation has lent itself to permitting a number of technical and experimental approaches that have addressed a range of novel scientific questions and answers not possible in other contemporary preparations.

## What is the WHBP?

The WHBP is a decorticate or decerebrate mouse or rat on full cardio-pulmonary bypass perfused arterially with cell-free Ringer solution containing an oncotic agent (polyvinylpyrrolidone, Ficoll, polyethylene glycol or sucrose) to prevent oedema. Decortication permitted studies of hypothalamic function *in situ* (Antunes et al., 2006). The original preparation contained the head, neck and thorax sectioned sub-diaphragmatically and perfused retrogradely via the descending aorta with an open circulation (veins were cut; Paton, 1996a). Subsequently, a fully intact rat preparation (a decerebrate, artificially perfused *in situ* preparation of rat; Pickering & Paton, 2006) was developed for studies on autonomic control of the bladder (Sadananda et al., 2011). The WHBP operates at 31°C, takes between 20 and 30 min to set up and depending on the experimental protocol remains viable for around 5 h. The preparation generates central respiratory drive and the pattern is critical for defining its viability (Fig. 1). The emergence and maintenance of a biphasic vagal nerve discharge of inspiration and post-inspiration is a critical characteristic of eupnoea and a vital indicator of optimal brain perfusion, and should be used as a guide when teaching trainees how to tune the WHBP during the critical initial phases of re-perfusion and re-oxygenation of the brainstem. To date, mice, rats, shrews (Smith et al., 2001) and guinea pigs (Dutschmann et al., 2019) have all been used and this has included neonates (all species), juveniles (rats) and adults

(shrews and mice). The Supporting information includes photographs and technical drawings of all the equipment needed to set up the WHBP.

## Some limitations of the WHBP

Some experience and training is needed to obtain reliable robust preparations as portrayed by inspiratory–post-inspiratory activity recorded from the vagus nerve; this requires some time, patience and a good grounding in integrative physiology. The WHBP is hyperoxic as the perfusate is bubbled with carbogen (95% $O_2$, 5% $CO_2$; Wilson et al., 2001) unless lower levels of oxygen are used (e.g. 40%). Note that the preparation is normocapnic. The viability time is limited ($\sim$5 h). It is restricted to neonatal and juvenile rats and guinea pigs, ineffective in adult rats and guinea pigs ($>$150 g), but is viable in mice of all ages. Thus, using rat models of cardio-vascular/respiratory diseases has the disadvantage that the animals studied will be young whereas in humans these diseases occur later in life. The animal size limitation is related to the volume of perfusate needed to be pumped to satisfy oxygen requirements and the demand that arterial pressure remains within the physiological range; excessive hydrostatic pressure causes oedema. We acknowledge that arterial pressure is lower than in *in vivo* animals, arterial baroreceptors are mostly unloaded, arterial shear stress and peripheral arterial resistance will be different, and there is a loss of lower body sensory afferent activity as well as both renal function and an immune system. The preparation is devoid of all hormones unless added back into the perfusate. The respiratory rate is slow due to the lower temperature (typically 31°C) and absence of Hering–Breuer feedback unless phrenic-triggered mechanical ventilation is used (Harris & St-John, 2005). The frontal cortices and thalamic structures are removed unless anaesthetic is added to the perfusate (St John et al., 2006). The latter means that diurnal mechanisms cannot be studied but it is possible to induce the rapid eye movement sleep state with pontine injections of carbachol (Brandes et al., 2011). There is a reduction in contrast between tissue types (e.g. nerve and muscle) because of the colourless perfusate, which can make dissection of small nerves challenging.

## The value of the WHBP for teaching and training students

From experience, the WHBP will give a trainee a unique opportunity to visualize, understand and explore fundamental concepts of integrative physiology by combining studies in neuroscience, cardiovascular and respiratory physiology, biophysics, and cell and molecular biology. Trainees that adopt the preparation quickest, and maintain its viability longest, will be those with a good grasp of cardiovascular and respiratory physiology; meticulous attention to details such as perfusion circuit cleanliness, precise Ringer solution constituents and surgery all contribute to preparation success. Once established, WHBP researchers will glean a novel perspective of the vital parameters needed for healthy respiratory and cardiovascular/autonomic systems and their function.

Success with the technique requires the aspiring physiologist to take on the additional role of the anaesthesiologist or intensivist in order to iteratively and efficiently answer the crucial question – how can I maintain the vitality of this complex system so I can test my experimental hypothesis? The answer lies in careful attention to positioning, perfusion pressure management, targeted temperature control and respiratory gas tensions, thereby ensuring the best chance of experimental success. It is interesting to note that ability to maintain these homeostatic facets germane to the WHBP relates to the core questions in intensive care research today: what temperature should we aim for after cardiac arrest (Dankiewicz et al., 2021)? How should we manage perfusion pressure to best protect the injured brain (Aries et al., 2012)? And what oxygen tensions portend good outcome in the critically ill (Mackle et al., 2020)?

Like many aspects of biology, the link between WHBP and clinical medicine is a symbiosis – taking the role of the clinician improves the chance of WHBP experimental success and taking heed of experimental insights gained from the WHBP enhances the clinician's understanding of vital physiology. While intensive monitoring of the patient forms the basis of critical care, our ability to isolate key aspects of homeostasis is limited, unfortunately, in the clinical scenario. For example, with the WHBP we can isolate brainstem sympathetic nervous system activity and thereby appreciate the integral role of respiratory activity in sympathetically mediated maintenance of blood pressure. In the intensive care unit, we do not yet have the ability to accurately monitor these crucial respiratory–autonomic–cardiovascular reflexes; however, when we do, knowledge directly gleaned from the WHBP will enable us to understand what they mean for patients.

We propose that the WHBP is a valuable training tool for future generations of integrative physiologists and physicians working in intensive care units.

## Exemplars of advancements in science

The following highlights illustrate some of the many discoveries made with the WHBP. The findings have often been reported in different laboratories using the *in situ* preparation and in many cases ratified using *in vivo* models.

## Novel insights into mechanisms of respiratory rhythms

The fascinating enigma of how the brain relentlessly and robustly generates the first and the last breath of life and at least ∼23,040 breaths/day in between (or ∼158,400 breaths/day in rat) remains incompletely understood. Since described by Lumsden in 1923, three patterns of automatic ventilatory activity can be generated by mechanisms intrinsic to the pons and medulla: eupnoea, apneusis and gasping.

The WHBP was employed to address some fundamental questions:

- Are all three patterns variants of a single fundamental rhythm, generated in a single brainstem region, or can respiratory rhythms be generated in multiple regions?
- What is (are) the mechanism(s) of respiratory rhythm generation – that is, is it due to the discharge of intrinsic burster, pacemaker neurons, the interactions among neurons in a circuit, or a combination of both intrinsic discharges and circuits?
- How are the brainstem mechanisms generating eupnoea, apneusis and gasping interrelated? How is eupnoea dominant and the others suppressed? How is eupnoea suppressed and apneusis and/or gasping released?

The WHBP was able to address these fundamental questions, as it fulfilled multiple criteria with the ability to:

- generate the three phase patterns of respiration (St John & Paton, 2000);
- sustain perturbations to allow switching between the respiratory rhythms repeatably and reversibly (St John et al., 2009);
- maintain preparation viability after the administration of pharmacological agents to establish the role of ionic channels and synaptic mechanisms (Paton et al., 2006).

The following sections address these questions in relation to the WHBP. The WHBP enabled those in the field of central respiratory control to address major controversies of the time.

**Gasping – nature's autoresuscitation mechanism.** In a series of studies, the *in situ* preparation allowed characterization of neuronal mechanisms by which eupnoea is suppressed and medullary mechanisms generating gasping are released (Paton et al., 2006; St-John et al., 2009). Gasping was generated by intrinsic neuronal mechanisms within the pre-Bötzinger complex (pre-BötC) whereas ponto-medullary neuronal circuits were essential for the generation and/or expression of eupnoea.

Gasping was generated by the intrinsic discharge of pacemaker–burster neurons. Whole cell recordings of neuronal activities in the ventro-lateral medulla indicated that these neurones have a discharge which commenced prior to, or during, the phrenic burst of eupnoea, discharged during the decrementing phrenic discharge of gasping induced by severe hypoxia, and were glutamatergic (Fig. 2). A limited population of these neurones, primarily those having expiration–inspiratory discharges in eupnoea, persisted with a periodic burster discharge after a total blockade of fast synaptic transmission by administration of antagonists of both inhibitory and excitatory synaptic transmission to the perfusate, which eliminated phrenic discharge (Fig. 2). These burster activities were eliminated by a blockade of persistent sodium channels, which also eliminated gasping, but not eupnoea, of intact *in situ* preparations.

In sum, the *in situ* preparation has enabled a direct testing of hypotheses concerning the mechanisms by which the respiratory pattern is switched from eupnoea to gasping. Such studies would have been almost impossible *in vivo*.

**Unique opportunities to define the hierarchical organization of brainstem circuits and the role of synaptic inhibition for the generation of respiratory rhythms.** A fundamental problem for understanding respiratory neural control is the spatial and functional organisation of brainstem respiratory networks. This organisation was studied using sequential rostral-to-caudal brainstem transections in the WHBP (Rybak et al., 2007; Smith et al., 2007). This approach was enabled by readily regulating perfusion pressure and perfusate gas composition with the *in situ* preparation. These studies tested hypotheses about the structural–functional organization of pontine–medullary respiratory networks proposing hierarchically organized functional compartments arranged bilaterally and rostro-caudally. The sequential sectioning approach was inspired by the micro-sectioning experiments of an *in vitro* neonatal rat brainstem–spinal cord preparation that resulted in the discovery of the pre-BötC (Smith et al., 1991; Fig. 3). A technical innovation designed for the WHBP was a specialized micro-translatable vibratome for bilateral serial micro-sectioning of the neuroaxis with high spatial precision while recording cranial, spinal respiratory motor outflows and regional respiratory population activities simultaneously.

These experiments revealed how brainstem circuits are spatially and functionally organised to generate the full complement of inspiratory–expiratory motor patterns in the mature nervous system. The major results were (see also Fig. 3):

- generation of the typical three-phase pattern required the presence of the pons providing excitatory drive to the ventral medullary respiratory column;
- a two-phase rhythmic pattern was generated without the pons when the Bötzinger complex (BötC) and pre-BötC were intact, which required excitatory drive from the retrotrapezoid nucleus (RTN);
- a one-phase rhythmic pattern was generated within the pre-BötC involving local endogenous mechanisms (see below).

These novel data drove the development of biophysical computational models of the core circuits within the brainstem respiratory network (Rubin et al., 2009; Smith et al., 2007) that were constructed to reproduce the experimental observations described above. In turn, these models led to predictions of plausible mechanisms essential for rhythmogenesis under various conditions,

one of which related to the intrinsic mechanisms of the neurones themselves.

**Intrinsic mechanisms essential for rhythmogenesis.** Debates about mechanisms of rhythm generation within the pre-BötC inspiratory oscillator have centred on cellular-level and synaptic biophysical mechanisms operating in excitatory (glutamatergic) neurons involving $Na^+$- and $Ca^{2+}$-based mechanisms (Jasinski et al., 2013). These mechanisms were proposed to explain the auto-rhythmic properties of pre-BötC neurons, and circuits. The $Na^+$-based mechanisms involve a slowly inactivating persistent sodium current ($I_{NaP}$) (e.g. Koizumi & Smith, 2008; Paton et al., 2006). A $Ca^{2+}$-activated non-selective cation current ($I_{CAN}$) coupled to intracellular calcium dynamics was also postulated (e.g. Phillips et al., 2019). In the WHBP, rhythm generation was disrupted by blocking $I_{NaP}$ but only when the network generated a one-phase rhythmic pattern analogous to that generated *in vitro*,

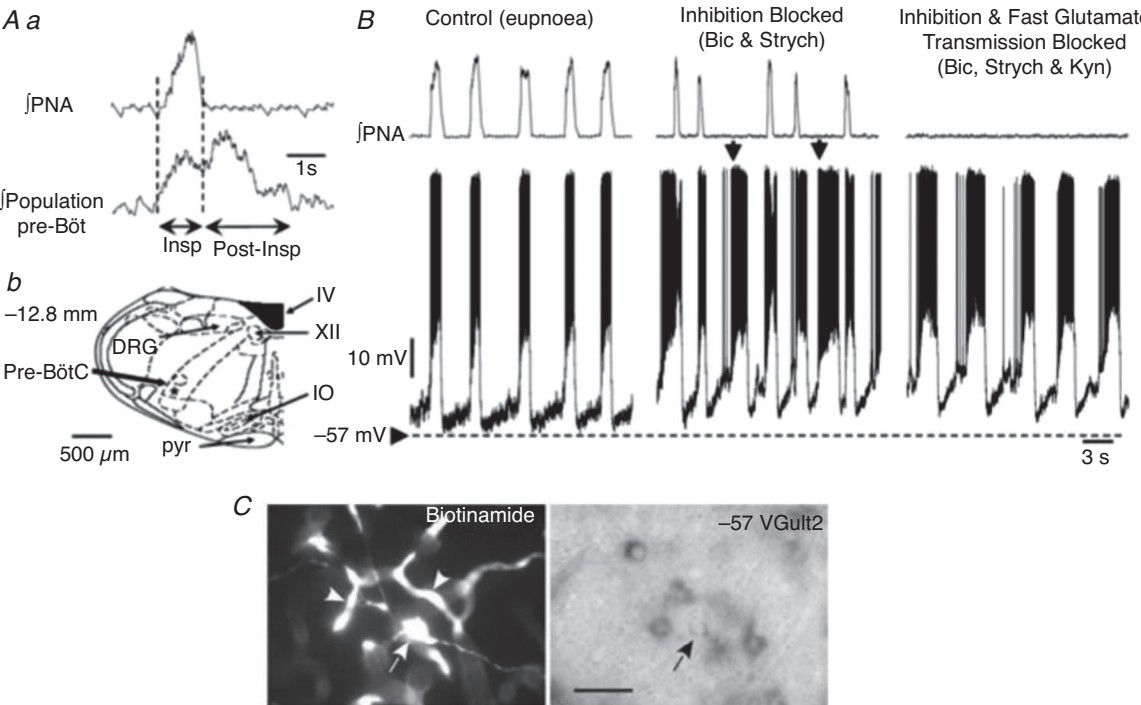

**Figure 2. A burster neurone recorded using a patch pipette from the ventral respiratory column**
Prior to patch recording, low impedance microelectrodes were used to map the respiratory column (*Aa*). The location of recording sites is depicted in *Ab*. Once mapped, patch pipettes were placed into this region and recordings made from inspiratory neurones during eupnoea (*B*). Note that membrane potential depolarized during neural expiration, with the discharge phase-locked with the phrenic bursts (PNA; *B*). Following addition of bicuculline (Bic) and strychnine (Strych) to the perfusate (inhibition blocked; *B*), neuronal bursting showed some uncoupling from the phrenic burst (arrows). Endogenous bursting was revealed on blocking both inhibitory and excitatory (kynurenic acid, Kyn) fast synaptic transmission – inhibition and fast glutamate transmission blocked. Note the absence of any phrenic discharge but phasic firing of intrinsic burster. *C*, photomicrographs of the burster neurone located in the pre-BötC. Its cell body (arrow) and proximal dendrites are shown in the left panel (biotinamide revealed with Alexa 488 fluorescence). Many capillaries located in the vicinity of the labelled neuron also contain biotinamide (arrowheads). The right panel shows the reaction product for VGLUT2 mRNA. Scale bar, 50 $\mu$m. Data from St-John et al. (2009). Abbreviations: DRG, dorsal respiratory group; IO, inferior olive; PNA, phrenic nerve activity; pyr, pyramidal tract, IV, fourth ventricle; XII, hypoglossal motor nucleus.

but not with the three-phase or two-phase rhythmic patterns (Smith et al., 2007), indicating state-dependence of mechanisms. Inhibitors of TRPM4/$I_{CAN}$ reduced inspiratory and post-inspiratory activities without disrupting rhythm generation (Koizumi et al., 2018). These results *in situ* gave major insights into the biophysical mechanisms contributing to amplitudes, patterns and frequency of respiratory rhythmic activity in the mature rodent system, which extended results from previous *in vitro* studies.

**Active expiration and an expiratory oscillator.** Another state of the respiratory network is when active expiration is recruited, which led to the idea of an interacting expiratory–inspiratory oscillator. Active expiration arises in response to exercise, hypercapnia, acidosis and hypoxia as a homeostatic response to increased ventilation. The *in situ* approach allowed interrogation of the role of an inhibitory connectome for this oscillation while maintaining blood gases clamped and without confounding cardiovascular responses. Data from neonatal and juvenile rats revealed expiratory neurones within the parafacial respiratory group (pFRG), which were silent under eucapnia and normoxia. Their activity emerged in phase with active expiratory motor output under hypercapnia, hypoxia or peripheral chemoreceptor activation (Abdala et al., 2009). Although this emerging oscillatory mechanism modulated other respiratory motor outputs (including hypoglossal, vagal and phrenic), it was not necessary for eupnoea at rest.

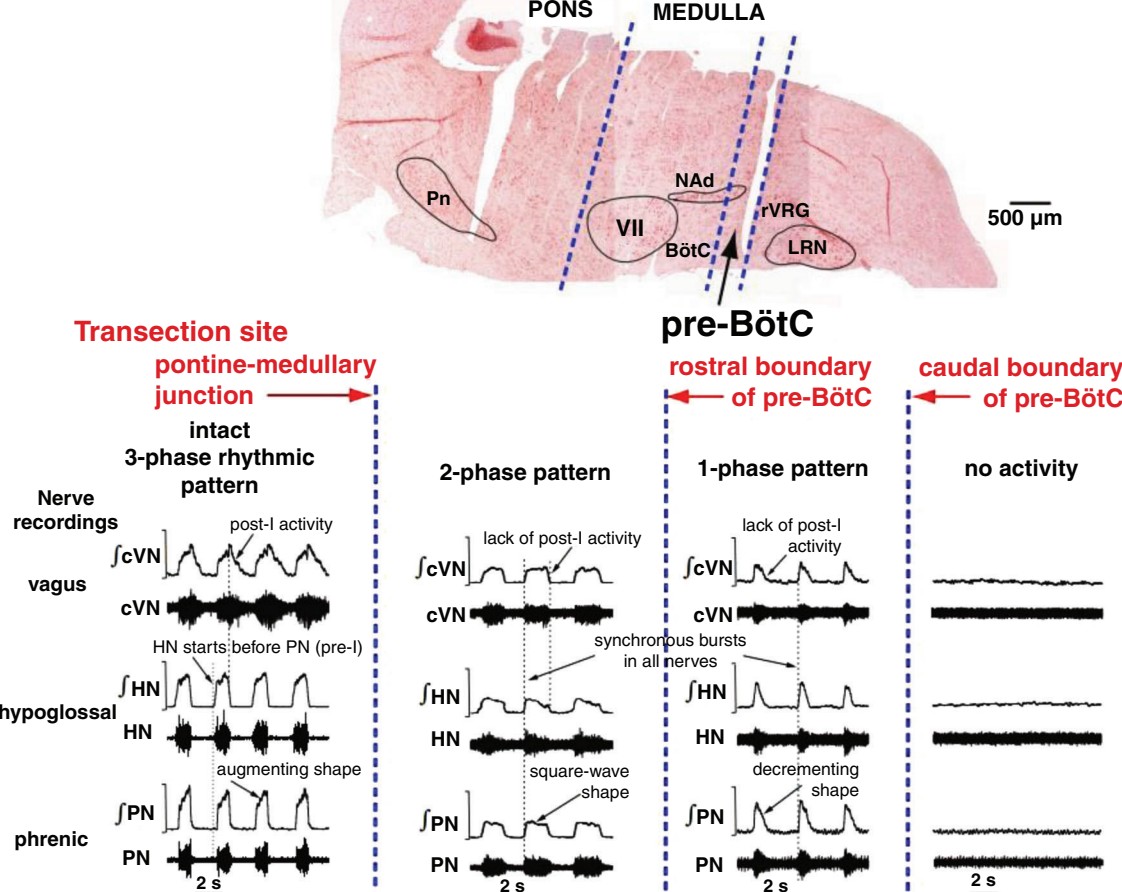

**Figure 3. Hierarchical rostro-caudal spatial organisation of the brainstem respiratory network**
Top, a sagittal section of the brainstem after transverse serial transections (note cut lines). Dashed lines indicate the transections at the pontine–medullary junction, the rostral boundary of the pre-BötC, and the caudal boundary of the pre-BötC. Bottom, the respiratory rhythm recorded from the central end of the vagus nerve (cVN; inspiratory and post-inspiratory discharge), the hypoglossal nerve (HN; pre-inspiratory and inspiratory) and the phrenic nerve (PN; inspiratory ramp) before (three-phase) and after (two-phase) ponto-medullary transection. The resulting two-phase pattern has no post-inspiratory or pre-inspiratory activities. The rhythm that exists with a transection at the rostral extent of the pre-BötC is a single-phase decrementing inspiratory pattern that is phase locked across all motor outputs and is abolished with sectioning at the caudal end of the pre-BötC. Abbreviations: LRN, lateral reticular nucleus; NAd, nucleus ambiguous; Pn, pontine nucleus; rVGRG, rostral ventral respiratory group; VII, facial nucleus. Data from Smith et al. (2007).

Moreover, the *in situ* microsection approach described above revealed that intact pontine connectivity was required for the emergence of this expiratory oscillator (Abdala et al., 2009). These *in situ* data motivated the expansion of computational models (Molkov et al., 2010; Rubin et al., 2011).

Further *in situ* data revealed that this expiratory oscillator was elicited by pharmacological disinhibition of the pFRG (de Britto & Moraes, 2017; Flor et al., 2020; Molkov et al., 2010) and that activation of Phox2b-expressing cells in this region was essential for its emergence under hypercapnia (Marina et al., 2010), although the expiratory oscillator did not express Phox2b or $CO_2/[H^+]$-sensitive ion channels/receptors (de Britto & Moraes, 2017; Magalhães et al., 2021). The conclusion reached was that the expiratory oscillator is inhibited by BötC/pre-BötC circuits during rest but emerges when excitatory drive from central chemoreceptors (located within the pFRG) and pFRG disinhibition occurs during hypercapnia or through disinhibition during hypoxia. Subsequent studies using opto- and chemo-genetic approaches in anaesthetized rats confirmed these *in situ* data (Huckstepp et al., 2015; Pagliardini et al., 2011) reflecting the validity of the WHBP.

**Importance of synaptic inhibition for the respiratory rhythm.** Unresolved is the inhibitory connectome for respiratory pattern generation (Richter & Smith, 2014;

Smith et al., 2013). By taking advantage of the ability to access the ventral surface of the brainstem *in situ*, bilateral, site-targeted pharmacological inhibition of glycine and $GABA_A$ receptors within the pre-BötC or BötC caused large site-specific perturbations of the rhythm and disrupted the normal three-phase motor pattern, in some experiments, terminating rhythmic motor output (Marchenko et al., 2016). In adult transgenic mice expressing Channelrhodopsin-2 under the VGAT promoter, photo-stimulation of inhibitory neurons within specific respiratory circuits was studied (Ausborn et al., 2018). Photo-stimulation demonstrated that inhibitory circuits within the pre-BötC/BötC regions critically regulate rhythmogenesis and are essential for normal respiratory motor pattern generation (Fig. 4).

**Glycinergic inhibition – essential for upper airway function from birth.** Prior to the WHBP, the superfused (passively oxygenated) *in vitro* approaches, such as the rhythmic slice preparation (Smith et al., 1991) and the *en bloc* medullary brainstem spinal cord preparation (Suzue, 1984) of neonatal rodents, were being employed for understanding mechanisms of respiratory rhythm generation. These early data indicated an absence of post-inspiratory activity. The WHBP revealed that, as early as a few hours after birth, contrary to *in vitro* preparations, post-inspiration existed and therefore a three-phase respiratory motor pattern, akin to that in

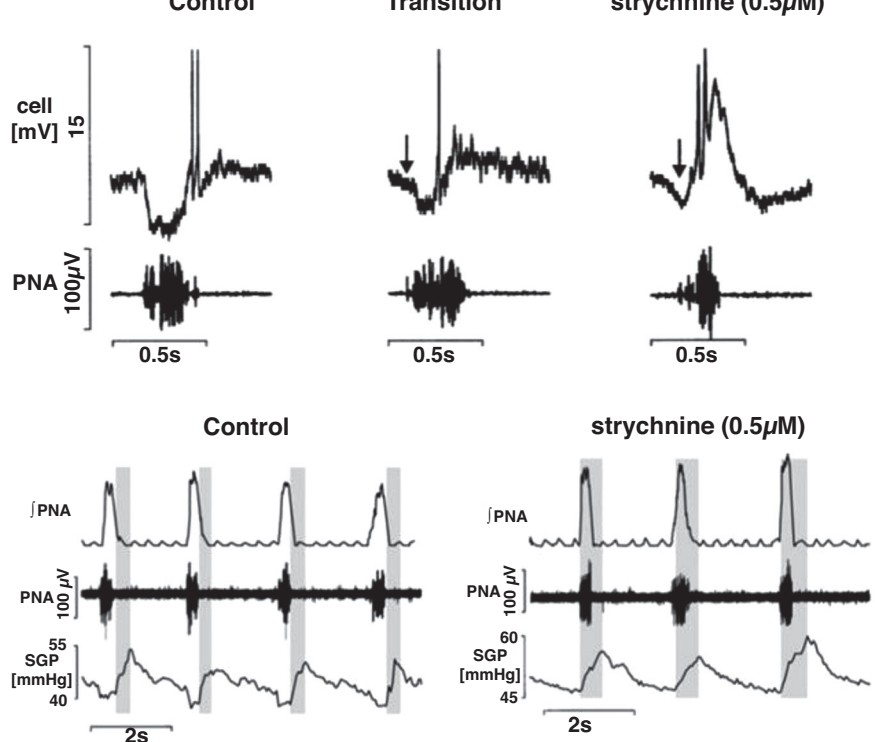

**Figure 4. Glycinergic synaptic inhibition was fundamental to the three-phase respiratory rhythm and control of the upper airway from birth**
Data from a 3-day-old rat showing post inspiratory activity (upper traces) characterised by inspiratory inhibition and post-inspiratory rebound excitation. Phrenic nerve activity (PNA) determined neural inspiration. In absence of glycinergic inhibitory synaptic transmission, the inspiratory hyperpolarisation reversed to depolarisation and firing. The effects on upper airway control were dramatic as subglottal pressure (SGP) (increase in pressure = glottal abduction, decrease = glottal adduction) increased during neural inspiration indicating elevated resistance, which is pathological. Glycine-mediated synaptic inhibition is vital for breathing and glottal coordination such as during swallowing. Data from Dutschmann and Paton (2002b).

adult mammals was present (Dutschmann et al., 2000; Fig. 4). The WHBP has provided the most detailed description of the features of the neonatal respiratory network in a mammal to date.

The *in vitro* data at the time suggested that glycinergic inhibition may have little to no function in central control of respiration in neonates, but instead may become more significant with maturation (Funk & Feldman, 1995; Richter & Spyer, 2001), In stark contrast, the WHBP demonstrated that glycinergic inhibition was essential for respiratory pattern formation in neonates (Dutschmann & Paton, 2002b). The first study that used the WHBP to investigate the consequence of a developmental loss of glycinergic inhibition in a transgene mouse line took place in the laboratory of Professor Diethelm Richter and showed that an absence of glycinergic inhibition triggered a shift of discharge onset of post-inspiratory neurons into the inspiratory phase (Büsselberg et al., 2001). This shift in the discharge caused a functional pathological glottal *constriction* during inspiration (Dutschmann & Paton, 2002b; Fig. 4). The conclusion was that glycinergic inhibition within the respiratory network is essential for post-inspiration and the phasic timing of the oscillation between inspiratory and post-inspiratory motor activities. This occurs at the earliest neonatal stages (Dutschmann and Paton, 2002a) and is essential for post-inspiratory behaviours in the newborn such as suckling and swallowing (Fig. 4).

**Perturbed excitation–inhibition balance causes respiratory disease.** Rett syndrome (RS) is a neurological disease in the autism spectrum that is accompanied by breathing disorders. Studies of transgene mouse models of RS suggested that an imbalance of synaptic inhibition and excitation may cause pathophysiological alterations in the functions of the pre-BötC (Medrihan et al., 2008) and Kölliker–Fuse nucleus (Stettner et al., 2007). The idea further advanced by Professor John Bissonnette was that, contrary to forebrain circuits, a lack of synaptic inhibition rather than too much excitatory synaptic transmission within the brainstem respiratory network may contribute to breathing irregularities in Rett syndrome. Using the WHBP of mutant mice deficient in methyl-CpG-binding protein 2 (MeCP2), respiratory irregularity and apnoea was associated with augmented post-inspiratory activity (Abdala et al., 2010; Stettner et al., 2007). Studies identified a deficiency of GABA perisomatic bouton-like puncta in the Kölliker–Fuse nucleus of Mecp2 heterozygous mice whereas blockade of GABA reuptake (to enhance GABA transmission) in the Kölliker–Fuse nucleus resolved breathing irregularity (Fig. 5). Conversely, blockade of $GABA_A$ receptors in this pontine region of healthy rats mimicked the Rett respiratory phenotype of recurrent central apnoeas with prolonged post-inspiratory activity (Abdala et al., 2016; Fig. 5).

**Post-inspiration is generated by a distributed ponto-medullary network.** Because the WHBP lacks the depressant effect of anaesthesia that is encountered *in vivo*, the preparation generates a robust post-inspiratory motor pattern that is observed in vagal or recurrent laryngeal nerve recordings. This has made the WHBP an invaluable platform to investigate the central origins of post-inspiratory activity which controls the strength and pattern of expiratory airflow (see Dutschmann et al., 2014). Initial studies identified that post-inspiratory motor activity is centrally gated by the Kölliker–Fuse nucleus (Dutschmann & Herbert, 2006), the key output node of the pontine respiratory group. More recently, studies using local perturbation of excitation–inhibition balance revealed that post-inspiratory activity might be controlled by synaptic interactions within a larger anatomically distributed ponto-medullary network (Dhingra et al., 2019).

Understanding the computational mechanisms underlying respiratory pattern formation across the ponto-medullary respiratory network implicitly requires techniques to monitor the population activity of the network at scale. To this end, the WHBP offers a key advantage over *in vivo* approaches because the dorsal brainstem surface can be fully exposed via cerebellectomy enabling highly invasive and repeated positioning of one or many multi-electrode arrays. A recent study leveraged this advantage to record respiratory local field potentials (rLFPs) at defined anatomical locations across the volume of the ponto-medullary brainstem (Dhingra et al., 2020). The resulting neuroanatomical 4-D maps of inspiratory, post-inspiratory and expiratory rLFPs (Supplementary Movie 1 (tjp14032-sup-0002-videoS1.avi) in Dhingra et al., 2020) revealed that population activity in the respiratory network peaks specifically and consistently at the transitions between respiratory phases. The anatomical localization of rLFPs showed, as predicted previously, that post-inspiration is the most distributed respiratory sub-circuit, engaging neuronal populations throughout the ponto-medullary network (Dhingra et al., 2020).

**Some limitations of studying respiratory rhythms with the WHBP.** We acknowledge that important aspects of the respiratory system are absent in the WHBP and these include pulmonary stretch receptor afferents activated on lung inflation, although they can be reintroduced using mechanical ventilation (Harris & St-John, 2005). Consequently, the respiratory rhythm generated is slow, which is also due to the hypothermia, whereas post-inspiratory activity is more abundant, typically prolonged because of increased expiratory time. Afferent feedbacks from the diaphragm, chest wall and airflow sensors are all absent. Although the preparation is devoid of the depressant effects of anaesthesia, it comes at a cost

of a loss of the forebrain and its influences on breathing, which in conscious rodents includes periods of aspiration, circadian rhythms and behavioural state (awake, sleep stages).

## Respiratory coupling to autonomic outflows

The WHBP has permitted novel functional insights into the respiratory coupling of autonomic outflows. Both sympathetic and cardiac parasympathetic outflows will be considered.

**Sympathetic nerve activity in the WHBP.** The observation of an increase in perfusion pressure during the initial stages of perfusion of the WHBP that coincides with re-commencement of respiratory activity was suggestive of restoration of sympathetic vasomotor activity and functional coupling between the systems. Subsequent sympathetic nerve activity (SNA) recordings from the

inferior cardiac nerve (Boscan et al., 2001), the thoracic chain (Simms et al., 2009), and internal and external cervical sympathetic branches (Roloff et al., 2018) have demonstrated a hexamethonium-sensitive SNA (Boscan et al., 2001) that is modulated robustly by baroreceptor or chemoreceptor reflex stimulation and is more congruent with SNA observations in conscious animals than the dampened reflex function of anaesthetised animals. Intracellular recordings from sympathetic pre-ganglionic neurones (SPN) have also been performed (Stalbovskiy et al., 2014; Fig. 6).

Although SPNs have been studied *in vitro* (e.g. Pickering et al., 1991; Yoshimura et al., 1986) and *in vivo* (Dembowsky et al., 1986; McLachlan & Hirst, 1980), these studies were limited by lack of reflex connectivity *in vitro* and low yields with intracellular recording *in vivo*. By adapting the WHBP, the first detailed study of the cellular properties and physiological activity of SPNs was made (Stalbovskiy et al., 2014). To achieve this, the piezoslicer discussed above (Smith et al., 2007) was used to

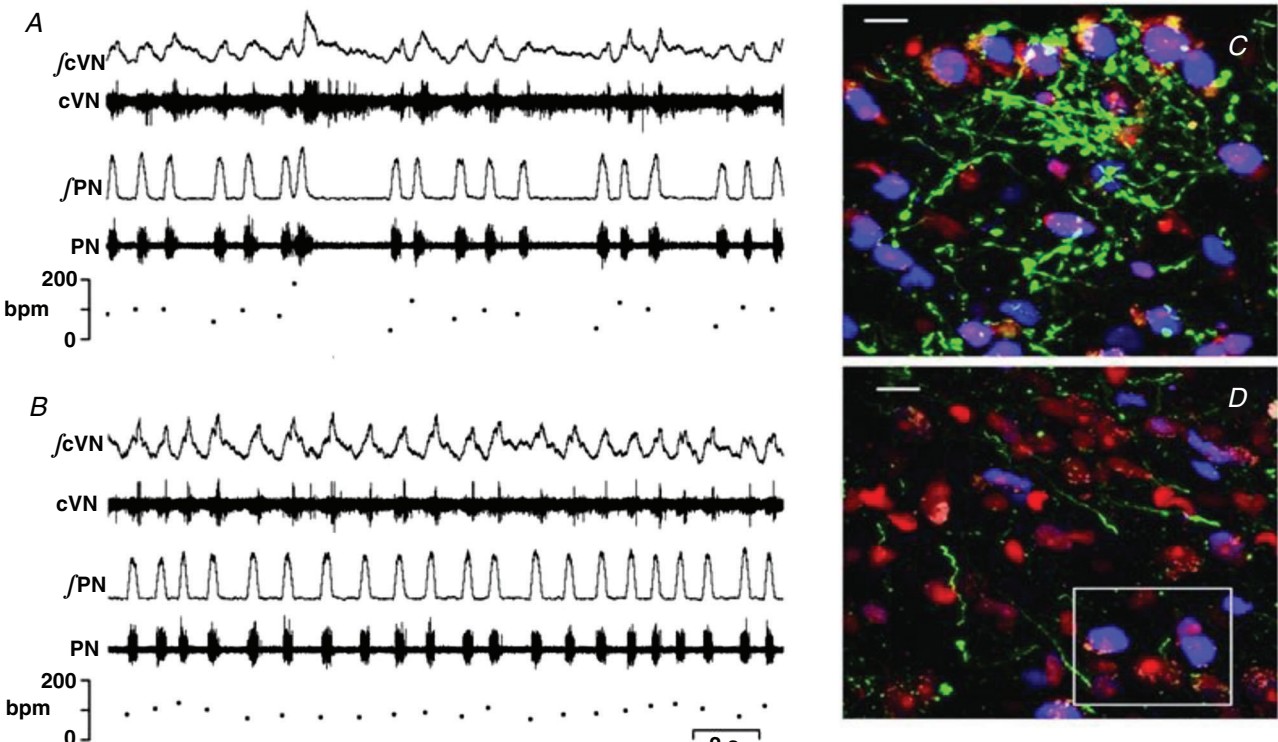

**Figure 5. Rescue of respiratory instability in a mouse model of Rett syndrome**
Original recordings of central vagus nerve (cVN), phrenic nerve (PN) and respiratory rate (breaths per minute, bpm) in a Mecp2$^{+/-}$ female mouse before (*A*) and after (*B*) microinjection of NO-711 (a GABA reuptake blocker, 10 $\mu$M, 60 nl) into the Kölliker–Fuse pontine nucleus. Note that the apnoeas are associated with prolonged post-inspiration as seen in cVN. These data emphasise that reduced synaptic GABAergic inhibition in the pons triggers respiratory instability. *C* and *D*, GABAergic neurones expressing eGFP under the control of the GAD67 promoter via a knock-in transgene (green); perikarya are labelled in red (Nissl stain) and MECP2 protein immuno-reactive (MECP2ir) nuclei are pseudo-coloured in blue within the Kölliker–Fuse nucleus. *C* is from a Mecp2$^{+/+}$/GAD67-eGFP female and *D* from a Mecp2$^{+/-}$/GAD67-eGFP littermate female. Note the marked reduction in GABAergic projections in the MECP2 deficient female mouse. Scale bars, 10 $\mu$m. Data from Abdala et al. (2016).

cut the thoracic spinal cord obliquely *in situ*. This allowed visualisation of the lateral horn and patch clamp access to SPNs with their central connections intact (Fig. 6). Their ongoing and reflex activity patterns classified them as muscle (MVC) or cutaneous (CVC) vasoconstrictor types, as described for sympathetic postganglionic fibre recordings *in vivo* (Janig, 2006). Key novel findings were:

- intrinsic properties of MVC and CVC type pre-ganglionic neurones were distinct;
- MVCs were powerfully inhibited at short latency (∼100 ms) by aortic baroreceptors (Fig. 6);
- CVC neurones were extensively coupled by gap junctions (as previously noted *in vitro*; Logan et al., 1996);
- MVC neurones were not coupled but had elevated intrinsic excitability in spontaneously hypertensive rats (SHR; a neurogenic model of hypertension) relative to

Wistar rat, which occurred before the development of hypertension (Briant et al., 2014).

**Sympathetic activity is coupled to, and driven, by respiration.** Respiratory modulation of SNA (Resp-SNA) is a hallmark of SNA, being first observed in the *in vivo* recordings of Adrian et al. (1932) and is evident in the WHBP at both pre- (Stalbovskiy et al., 2014) and post-ganglionic levels (Simms et al., 2009; Fig. 7). In the WHBP, Resp-SNA is enhanced in rodent models of disease (Fig. 7) including the SHR (Simms et al., 2009), chronic intermittent hypoxia (CIH; Zoccal et al., 2008), a maternal protein restriction model (de Brito Alves et al., 2015), a uteroplacental insufficiency model (Menuet et al., 2016), two kidney, one clip model of renal hypertensive rats (Oliveira-Sales et al., 2016), and a genetic model of rat chronic kidney disease (Saha et al., 2019). The respiratory phase of modulation of SNA is disease and target organ specific: for example, in Wistar rats, thoracic SNA peaked

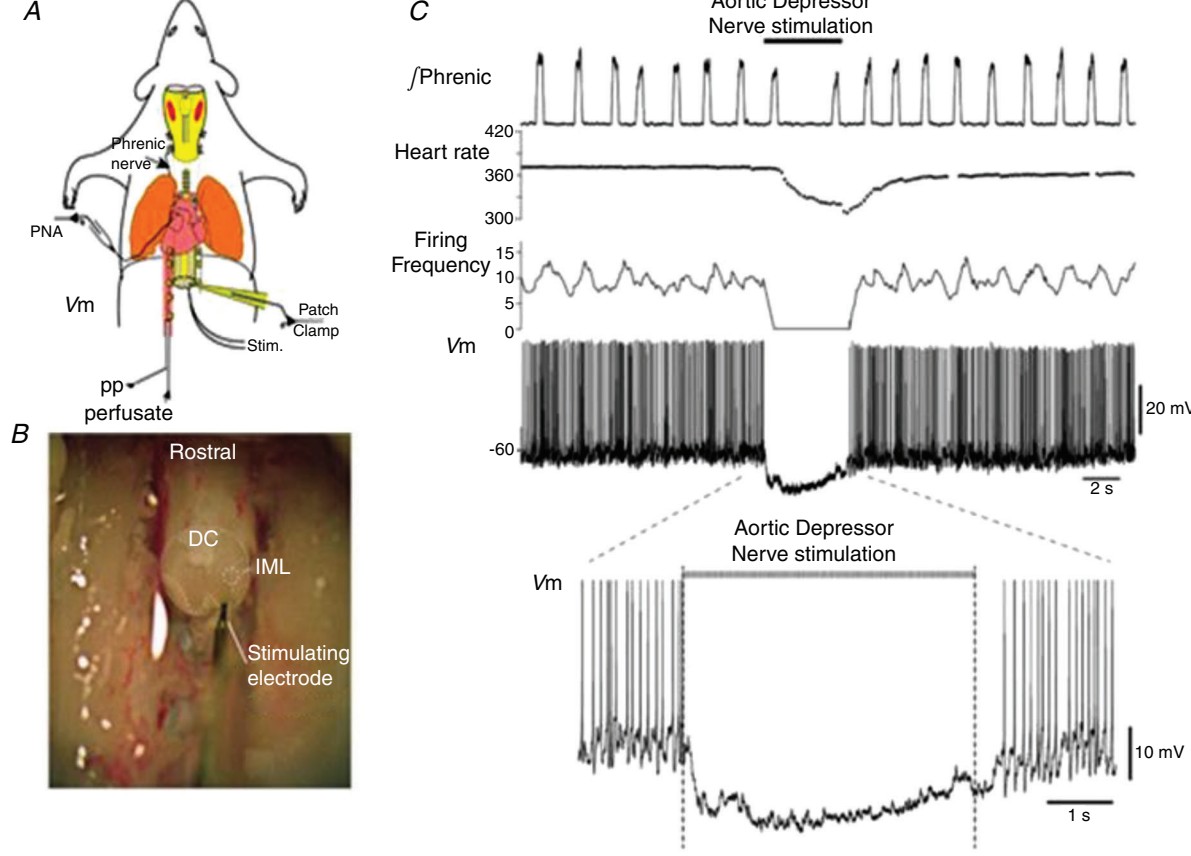

**Figure 6. Whole cell recordings of functionally identified pre-ganglionic sympathetic recordings in neonatal rats**

*A*, schematic WHBP of neonatal rat. Phrenic nerve and patch clamp recordings from sympathetic pre-ganglionic neurones (SPN) were made. *B*, thoracic spinal cord is exposed and an oblique cut through the cord was made to visualise inter-mediolateral (IML) cell column. SPNs were antidromically activated by stimulating the ventral root. DC, dorsal columns. *C*, electrical stimulation of the aortic depressor nerve hyperpolarised SPNs and caused a marked bradycardia. This baroreceptor inhibition was of short latency (within 100 ms). Data from Stalbovskiy et al. (2014).

across the inspiratory–post-inspiratory transition but is shifted into the inspiratory phase in the SHR (Menuet et al., 2017; Simms et al., 2009; Fig. 7) as observed in the renal nerve *in vivo* previously (Czyzyk-Krzeska & Trzebski, 1990). Whilst an increase in inspiratory-related SNA was also reported in the CIH model (Zoccal et al., 2019), there was also an additional peak in late expiration (Zoccal et al., 2008), also seen in the cervical sympathetic

branch of the SHR (Roloff et al., 2018). In the SHR, the elevated SNA occurred prior to the development of hypertension, pointing to a causal role for SNA in developing the hypertension (Simms et al., 2009).

Whilst patterns of Resp-SNA differed between models of hypertension, the key outcome is the same – exaggerated respiratory-related elevations in blood pressure (Fig. 7). These modulations are Traube–Hering

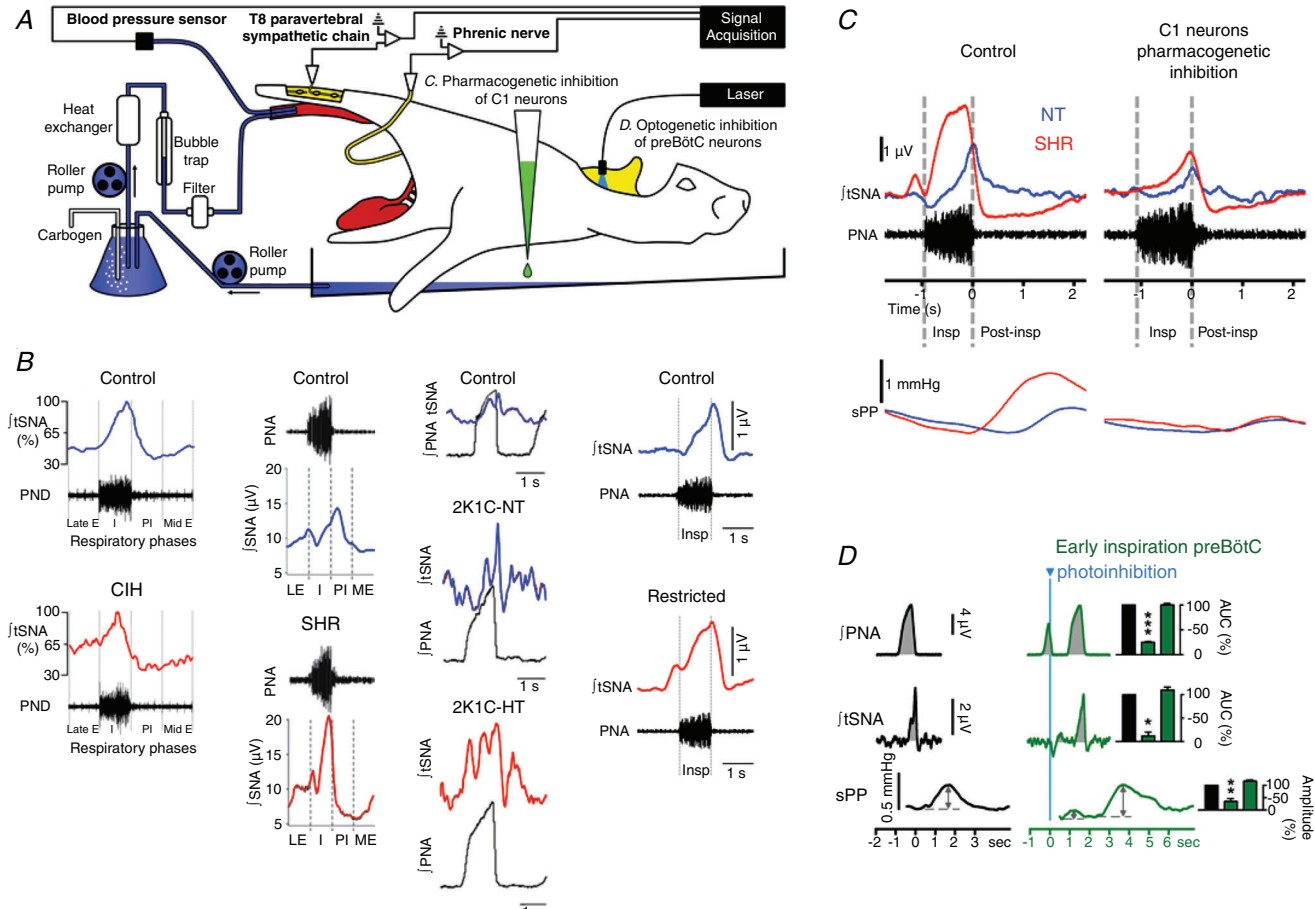

**Figure 7. The WHBP enables recordings of tSNA, showing enlarged inspiratory Resp-SNA in models of hypertension, likely generated via inspiratory modulation of C1 neurons by pre-BötC neurons**

*A*, recordings of thoracic (T8–T10) vasomotor sympathetic nerve activity (tSNA) in the WHBP, at the level of the paravertebral sympathetic chain. *B*, when compared to phrenic nerve activity (PNA), tSNA shows phasic respiratory-locked activity (Resp-SNA), principally during inspiration, with a peak at the inspiration–post-inspiration phase transition, as seen in all recordings of control animals (blue). In chronic intermittent hypoxia (CIH) Resp-SNA (%) is amplified during late expiration. However, in spontaneously hypertensive rat (SHR), 2 kidney 1 clip (2K1C) and uteroplacental insufficiency (restricted) rat models of hypertension, Resp-SNA is amplified during inspiration with a peak shift into inspiration (red). In most models, the exaggerated Resp-SNA appears before the development of hypertension. This is highlighted on the 2K1C-Normotensive (NT) trace, which shows exaggerated Resp-SNA during inspiration compared to control, even though the animal is normotensive at this stage. Further increases occur at the hypertensive stage (2K1C-HT). *C*, pharmacogenetic inhibition of C1 neurons in the WHBP, as shown in *A* by adding the ligand to the perfusate, decreased Resp-SNA of SHR (red) to normotensive (NT; blue) level, with a Resp-SNA peak that is shifted back to the inspiration–post-inspiration phase transition and decreased Traube–Hering waves in the SHR to NT level, as seen on systolic perfusion pressure (sPP). *D*, single-pulse optogenetic inhibition of pre-BötC neurons during early inspiration in the WHBP, as shown in *A* by placing optical fibres linked to a laser bilaterally in the medulla oblongata, stopped the on-going inspiratory burst, strongly decreasing the associated Resp-SNA and Traube–Hering wave. Adapted from data in Menuet et al. (2016, 2017, 2020), Oliveira-Sales et al. (2016), Simms et al. (2009), Zoccal et al. (2008).

waves. Using the WHBP with a hypocapnic perfusate, central respiratory activity ceases, giving a quantitative appreciation of the influence of Resp-SNA for generating arterial pressure. Because the decay in the Traube–Hering wave is relatively slow, the subsequent wave summates with it so driving pressure up. Traube–Hering waves were found to contribute ∼20 mmHg in the SHR (Simms et al., 2009).

**Central coupling mechanisms of Resp-SNA in the SHR.** A major source of excitatory drive to sympathetic outflow is from the rostral ventrolateral medulla oblongata (RVLM). Bulbospinal RVLM neurons show different patterns of respiratory modulation in the WHBP, with one group, the adrenergic C1 neurons, displaying inspiration-locked activation in Wistar rats (Moraes et al., 2013). In the SHR, inspiratory and post-inspiratory barosensitive, bulbospinal RVLM neurons show enhanced activity (Moraes, Machado et al., 2014). However, the unknowns are:

- What proportion of the Resp-SNA was dependent on C1 neurones?
- What was the precise origin of the respiratory drive(s) to C1 neurones?
- Was there a change in excitability within the respiratory and/or the C1 neurone(s)?

*Role of C1 neurones in Resp-SNA coupling.* Pharmacogenetic inhibition of C1 neurons, using the *Drosophila melanogaster* allatostatin receptor, which opens G-protein-coupled inward-rectifying potassium channels, induced a 30% decrease in Resp-SNA amplitude in normotensive WHBP rats (Marina et al., 2011). Using the same strategy to inhibit C1 neurons in WHBPs of SHR, Resp-SNA was reduced and the amplitude of the Traube–Hering waves normalised to normotensive rat level. The peak of Resp-SNA was also shifted back to the inspiratory–post-inspiratory transition as seen in normotensive rats (Menuet et al., 2017). These effects validated C1 neurons as a central node transmitting enhanced inspiratory-locked activity into vasomotor sympathetic activity in the SHR, to enhance Traube–Hering waves; these WHBP data were validated *in vivo* (Menuet et al., 2017).

*Origin of the respiratory drive(s) to C1 neurones.* Pre-BötC neurons were identified as a potential driver of the inspiratory-mediated modulation of C1 neurones (Menuet et al., 2020; Fig. 7). Whole cell recordings in the WHBP showed that SHR pre-BötC neurons have increased intrinsic excitability (Moraes, Machado et al., 2014; see below). Using optogenetic inhibition and excitation, pre-BötC neurons were found to be sympathoexcitatory, and photoinhibition during early or

late inspiration reduced Resp-SNA and Traube–Hering waves (Menuet et al., 2017, 2020; Fig. 7).

*Is the intrinsic excitability of respiratory and/or the C1 neurone(s) altered in the SHR?* After their physiological identification, neurones were studied in synaptic isolation, by adding blockers of fast inhibitory and excitatory transmission to the perfusate (Fig. 8). Using whole cell recording to study firing responses to injected current, post-inspiratory neurones of the SHR were found to be electrically more excitable than those in Wistar rats (Fig. 8); this was due to a reduction in the calcium modulated potassium channel conductance (BK type; Moraes, Machado et al., 2014; Fig. 8). In contrast, the electrical excitability of C1 neurones (but also RVLM non-C1 neurones) was not different between SHR and Wistar rats (Moraes, Bonagamba et al., 2014; Moraes, Machado et al., 2014). These results strongly suggested that sympathetic overactivity results from changes in membrane excitability of distinct ventral medullary respiratory neurones driving C1 cells. These data support the novel contention that the sympathetic hyperactivity in hypertension is caused by a change in the intrinsic activity of respiratory neurones.

**Enhanced Resp-SNA caused by central chemoreceptors and chronic intermittent hypoxia.** Hypercapnic stimulation of central chemoreceptors produced a unique respiratory related burst in sympathetic activity occurring during late expiration similar to that observed in rats treated with CIH, which causes sympathetically dependent hypertension (Molkov et al., 2011; Zoccal et al., 2007, 2008, 2009). This led to the discovery that CIH triggered persistent activation of the pFRG containing the expiratory oscillator (see above) and caused recruitment of abdominal motor activity mediating forced expiration (Machado et al., 2017; Molkov et al., 2011; Moraes et al., 2013). The emergence of late-expiratory SNA increased peripheral vascular resistance and elevated arterial pressure (Zoccal et al., 2018). Unlike data from the SHR (see above; Menuet et al., 2017), acute and selective chemogenetic inhibition of C1 neurones did not affect the enhanced sympathetic outflow of CIH rats (Moraes et al., 2017). Thus, sympathetic overactivity in CIH hypertension is not due to elevated intrinsic excitability of RVLM pre-sympathetic neurons but rather is dependent on elevated excitability of upstream respiratory (pFRG late-expiratory) neurones potentially projecting to non-C1 RVLM pre-sympathetic neurones. Thus, distinct respiratory–sympathetic network interactions generate apparently a similar functional outcome, namely neurogenic hypertension.

**Respiratory modulation of cardiac parasympathetic outflow in rats.** The presence of respiratory sinus

arrhythmia (RSA) is considered an important prognostic indicator of cardiovascular health, from good health with high RSA, to poor outcomes when the loss of RSA can indicate the likelihood of conditions such as sudden cardiac death. Transection between the pons and medulla in the WHBP resulted in the disappearance of RSA, indicating that pontomedullary connections were essential (Baekey et al., 2008; Farmer et al., 2016). However, such transection severely reduces excitatory inputs to cardiac vagal preganglionic neurons, masking potential inhibitory influences coming from within the medulla. In the WHBP (and *in vivo*) both a tonic and a respiratory modulated source of cardiac parasympathetic inhibitory tone come from a subset of pre-BötC neurons (Menuet et al., 2020). It is likely that RSA is shaped by the dual action of excitatory pontine Kölliker–Fuse post-inspiratory neurons, and inhibitory pre-BötC expiratory and inspiratory neurons.

The WHBP enabled the first intracellular recordings from vagal ganglion cells on the surface of the beating atrium, whilst maintaining physiological preganglionic inputs from the brainstem (McAllen et al., 2011; Fig. 9). This was possible in the WHBP because the ventricles could be cut away and the atria laid open and pinned to a foot plate for stability (Fig. 9). The bloodless field allowed visualisation of the vagal ganglia. Two types of neuron were encountered:

- presumed interneurons that were silent under experimental conditions and showed phasic intrinsic membrane properties;
- principal neurons that received ongoing excitatory synaptic activity with the expected respiratory modulation peaking in late inspiration – early post-inspiration.

No IPSPs were detected (Fig. 9). Stimulations of baroreceptor, chemoreceptor and diving afferents all evoked a strong increase in synaptic activity coincident with bradycardia. Each ganglion cell received input from a single preganglionic neuron, which conveyed all tested tonic and reflex synaptic drive. Frequency-dependent depression reduced the amplitude of closely successive EPSPs (Fig. 9), and this resulted in a proportion of spike failures during periods of strong synaptic bombardment. We concluded that the vagal pre-to-post-ganglionic synapse acted not as an integrator

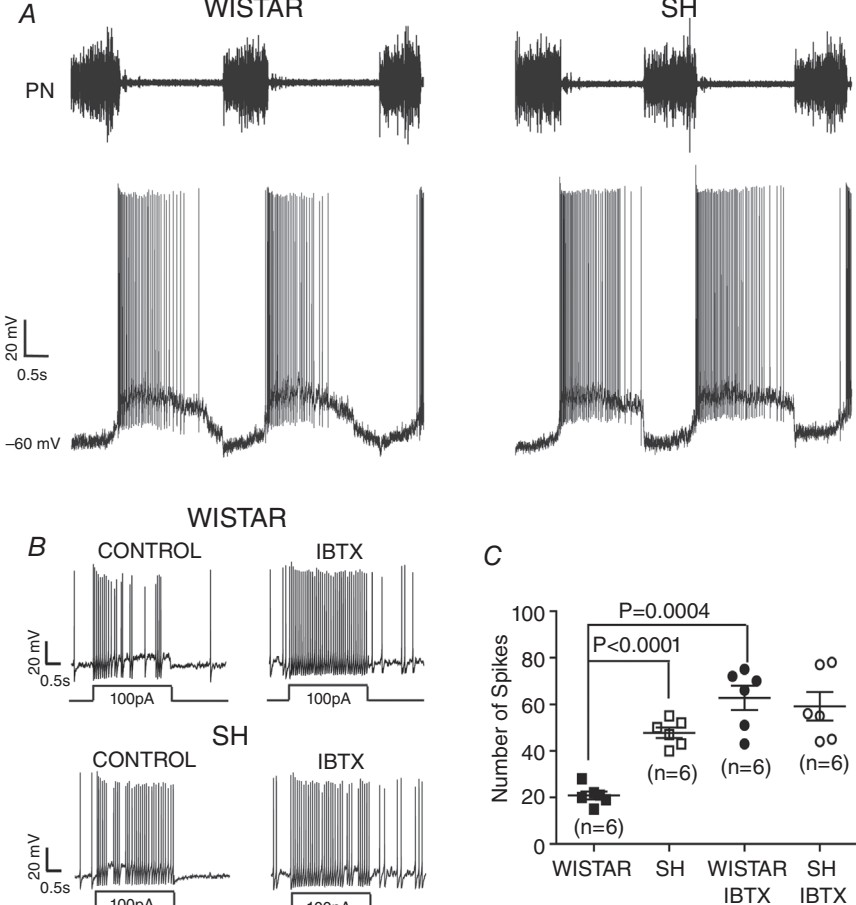

**Figure 8. Sympathetic overdrive in neurogenic hypertension is, in part, driven by the elevated electrical excitability of post-inspiratory neurones that connect to pre-sympathetic neurones in the rostral ventrolateral medulla**
*A*, the firing of post-inspiratory neurones in normotensive Wistar and spontaneously hypertensive (SH) rats shows inspiratory hyperpolarisation and a decrementing pattern of discharge; note the longer and more powerful burst discharge in the SH rat (arrowed). Once synaptically isolated by blocking fast excitatory and inhibitory transmission (by addition of kynurenate, bicuculline and strychnine), iberiotoxin (IBTX) blockade of the large conductance calcium-dependent potassium channels increased the electrical excitability of post-inspiratory neurones (to injection of depolarising current) in Wistar rat (Control, panel *B* (left)) to those seen in the SH rat (Control, panel *B* (right) and *C*).
Abbreviation: PN, phrenic nerve. Data from Moraes et al. (2014a).

but as a frequency-dependent gate of incoming synaptic traffic.

**Some limitations of studying respiratory coupling to autonomic outflows in the WHBP.** The absence of lung inflation will affect the respiratory rhythm and pattern (see above) so autonomic coupling may be affected, although it was similar to *in vivo* vagotomised rats (Czyzyk-Krzeska & Trzebski, 1990). If the perfusion pressure is set low (<60 mmHg), this will be sub-threshold for baroreceptor activation such that an absence of pulse pressure will prevent pulse-modulation of sympathetic discharge as seen *in vivo* in animals and humans. An absence of baroreceptor activity may also affect the respiratory rhythm as these sensors can influence the onset of inspiration (Barnett et al., 2021). The absence of some end organs makes it difficult to find specific sympathetic nerves (mesenteric, renal, splenic, for example). Thus, many recordings are from the sympathetic chain at different levels (cervical, thoracic or lumbar). This may limit studies on differential control of sympathetic activity when organ-specific outflows are required. However, this can be achieved in the whole rodent variants of the WHBP – the decerebrate arterially perfused rat where recordings can be made of renal and adrenal nerves (Simms et al., 2007) and the pelvic nerve (Ito et al., 2019).

## Reflex modulation of sympathetic and respiratory activity in the WHBP

**Salt-sensing WHBP.** An investigation of salt sensing mechanisms within the forebrain was undertaken by developing the working heart–brainstem–hypothalamus preparation (WHBHP). Whilst the WHBP was decerebrated pre-collicularly, the WHBHP was decorticated with the preoptic area, its adjacent septal nuclei and hypothalamus intact (Antunes et al., 2006). The challenge was to demonstrate that the hypothalamic neuronal circuitry was functional. Switching to a hyper-osmotic perfusate increased lumbar sympathetic nerve activity (LSNA) due to excitation of osmoreceptors located in the subfornical organ (SFO; a circumventricular organ), which activate presympathetic neurones located in the paraventricular nucleus of hypothalamus (PVN) which, in turn, activated sympathetic outflow. Significant increases in the levels of oxytocin and vasopressin were observed in the WHBHP perfusate (Pires da Silva et al., 2016). The SFO and its hypothalamic connectivity were

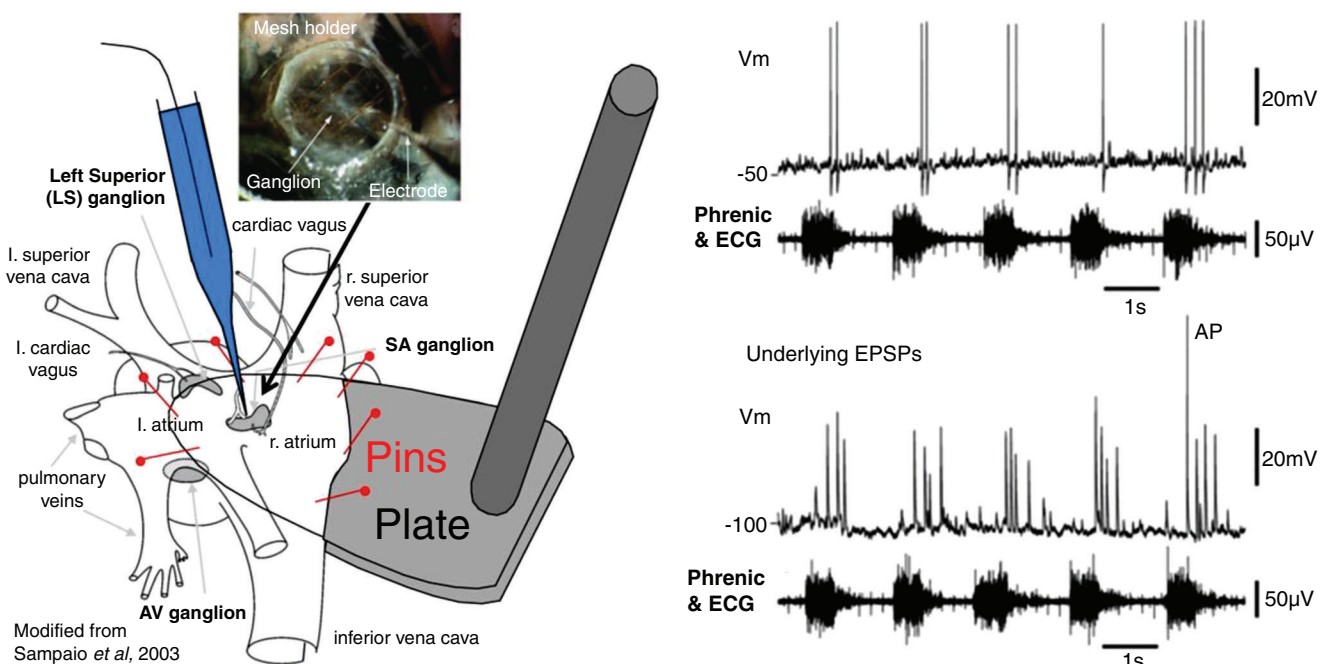

**Figure 9. The first intracellular recordings of post-ganglionic cardiac vagal neurones with central nervous connectivity preserved**
Left, using a WHBP where the cardiac ventricles have been removed and the right atrium slit open and pin mounted on a small plate, sharp microelectrode impalements were made from cardiac ganglion neurones. The atrium continued to beat (see ECG) and the ganglia were held steady using a fine nylon mesh (inset). Right, an example of the firing pattern of a cardiac post-ganglionic neurone; note it fired in post-inspiration (above) and these actions potentials were driven by large EPSPs that typically did not summate (bottom). Tests performed indicated a one-to-one connection of these neurones with a pre-ganglionic cardiac vagal motoneurone. Data from McAllen et al. (2011).

demonstrated by using a controllable microblade coupled to a 3-D micromanipulator that enabled connections between these structures to be severed before and after hyperosmotic stimulation. This eliminated the LSNA response proving salt-sensitive functionality of the SFO and the importance of connectivity with the PVN (Antunes et al., 2006).

With integrity confirmed, it was demonstrated for the first time that the hyperosmotically induced sympathoexcitation was mediated, primarily, by the PVN and its vasopressinergic spinal projections acting on V1a vasopressinergic receptors within the spinal cord (Antunes et al., 2006). Further, these osmotically evoked increases in sympathetic activity appear to be mediated by novel peptidergic and purinergic signalling within the PVN (Buttler et al., 2016; Ferreira-Neto et al., 2017; Ribeiro et al., 2015). In contrast, dehydration, another potent osmotic stress that leads to sympathoexcitation, was not dependent on the hypothalamus but the commissural nucleus of the tract solitarius (Colombari et al., 2011).

**Arterial baroreceptors – pressure-dependent recruitment of autonomic limbs.** Raising arterial pressure in the WHBP to stimulate baroreceptors highlights a pressure-threshold differential in the recruitment of sympathetic (withdrawal) *vs.* parasympathetic (activation/bradycardia) of the baroreflex (Fig. 10). The non-cardiac SNA baroreflex had a threshold of 66 *vs.* 82 mmHg for the cardiac SNA baroreflex (Simms et al., 2007). Likewise, the cardiac parasympathetic baroreflex component was also only active over a higher pressure range (Fig. 10). Thus, there is a hierarchical recruitment of the autonomic output limbs of the baroreflex with a sympathetic predominance at lower arterial pressures.

Further studies pointed to a dominant role of the aortic baroreceptors for mediating the baroreflex bradycardia to a pressure stimulus (Pickering et al., 2008).

**Peripheral chemoreceptors – purinergic signalling essential for sympathetic activation.** With an absence of anaesthesia and lung inflation, the primary peripheral chemoreflex response of hyperpnoea, hypertension, bradycardia, raised sympathetic activity and heightened central respiratory drive is vibrant in the WHBP (Fig. 11). One discovery was the finding of both hyperreflexia and hypertonicity of carotid body afferent discharge in SHR *vs.* Wistar rats. Subsequently, carotid body afferents were identified as drivers of the elevated Resp-SNA in SHR (McBryde et al., 2013) due to upregulation of purine P2X3 receptors in chemoreceptive petrosal neurones, as identified using single cell qRT-PCR (Pijacka et al., 2016). In WHBP of SHR, petrosal chemoreceptive neurones are exquisitely sensitive to ATP (Moraes et al., 2018; Fig. 11). Blockade of P2X3 receptors confined to the carotid body lowered the sensitivity of chemoreceptive petrosal neurones (Fig. 11), as well as sympathetic activity by normalising the Resp-SNA coupling in SHR. Thus, the carotid body afferent activity plays a major role in the exaggerated Resp-SNA coupling in SHR. As predicted from the WHBP studies, blocking P2X3 receptors systemically lowers blood pressure and normalises carotid body sympathetic reflex sensitivity *in vivo* (Pijacka et al., 2016). It is noteworthy that carotid body chemoreflex-evoked sympathoexcitation is also dependent on P2 receptor transmission within the nucleus of the solitary tract (Braga, Soriano et al., 2007; Paton et al., 2002). Thus, purinergic transmission is a specific and selective marker of carotid body-mediated sympathoexcitation (Zera et al., 2019).

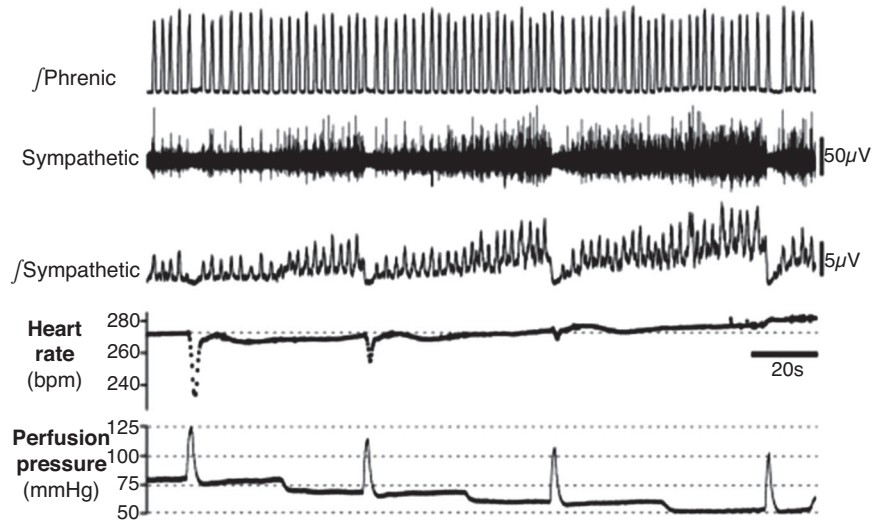

**Figure 10. Differential pressure sensitivity of baroreflex autonomic limbs**
The heart rate and sympathetic nerve responses to pressor challenges at different baseline pressures indicates that the baroreflex bradycardia occurs at higher pressure threshold than the sympathoinhibition. Note that the progressive increase in the sympathetic discharge with reductions in perfusion pressure with little change in basal heart rate. Data from Simms et al. (2007).

**Isolating peripheral, brainstem and spinal chemoreceptors in WHBP variants.** The juvenile and neonatal arterially perfused WHBP is well-oxygenated and only slightly acidic under baseline conditions (Dutschmann et al., 2000; Wilson et al., 2001). Moreover, as in intact animals, reducing perfusate $P_{CO_2}$ of the WHBP causes cessation of respiratory activity (i.e. apnoea), while reducing perfusate $P_{O_2}$ causes a transition from eupnoea to gasp-like motor patterns. These properties, together with the WHBP being arterially perfused, have allowed independent characterization of neural outputs resulting from stimulation of peripheral, brainstem and spinal cord chemoreceptors in WHBP descendants (Fig. 12).

A dual perfused preparation (DPP; Day & Wilson, 2005, 2008; Fig. 12) was derived from the WHBP in which

two perfusion lines are used: one for the descending aorta supplying the CNS and pressure regulated by a proportional integral derivative (PID) feedback controller, and the other for the common carotid arteries (the common carotid arteries are each sutured shut proximal to the heart). This was the first *in situ* preparation that allowed normoxic perfusion ($P_{O_2}$ = 100 Torr) of the carotid bodies, independent of the hyperoxic brainstem perfusion ($P_{O_2}$ > 600 Torr), which would otherwise prevent normal carotid body glomus cell chemosensory activity.

Using the DPP, the interaction of the peripheral $O_2/CO_2$ and brainstem $CO_2$ respiratory chemoreceptors in rats was found to be hypoadditive (i.e. a system that has chemoreflex redundancy in which each chemoreflex alone

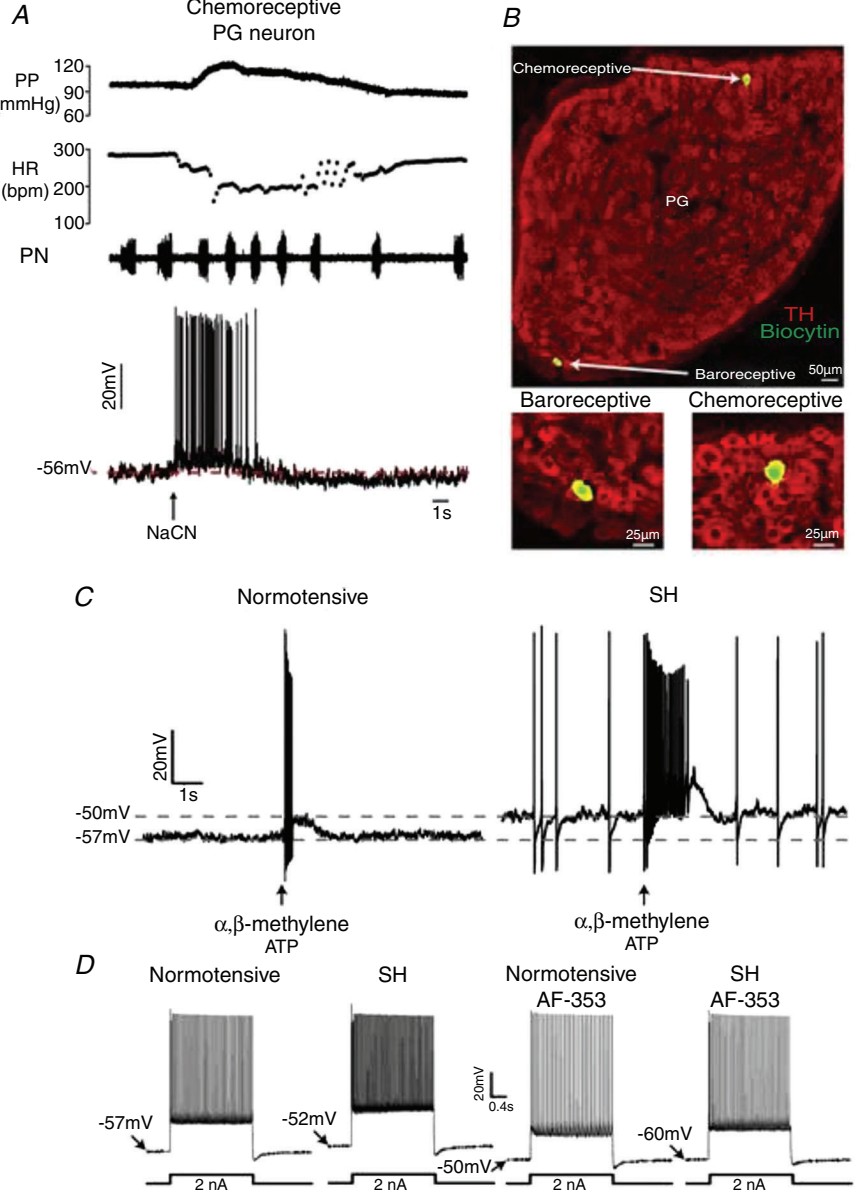

**Figure 11. Chemoreceptive petrosal neurones are electrically more excitable in hypertension**

*A*, whole cell recording from a petrosal neurone relaying inputs from the carotid body to the brain in the WHBP. Note that the reflex responses – increase in perfusion pressure (PP), decrease in heart rate (HR) and elevation in phrenic nerve (PN) discharge frequency evoked by sodium cyanide (NaCN) can be recorded simultaneously with the firing response. Single cell qRT-PCR is possible (not shown, see Pijacka et al., 2016) and neurone labelling (*B*); note locations of neurones relaying chemo- and baro-receptive information are distinct. *C*, in spontaneously hypertensive (SH) rats petrosal chemoreceptive neurones are more depolarised and exhibit ongoing firing and are more sensitive to the stable ATP analogue $\alpha$,$\beta$-methylene ATP pico-injected into the ipsilateral carotid body. *D*, the elevated electrical excitability of hypertensive petrosal chemoreceptive neurones is in most part mediated by purinergic P2X3 receptors: AF-353 (a P2X3–P2X2/3 receptor blocker) picoinjected into the ipsilateral carotid body hyperpolarises the membrane potential, abolishing ongoing firing and the repetitive firing response to injected current. Data from Moraes et al. (2018).

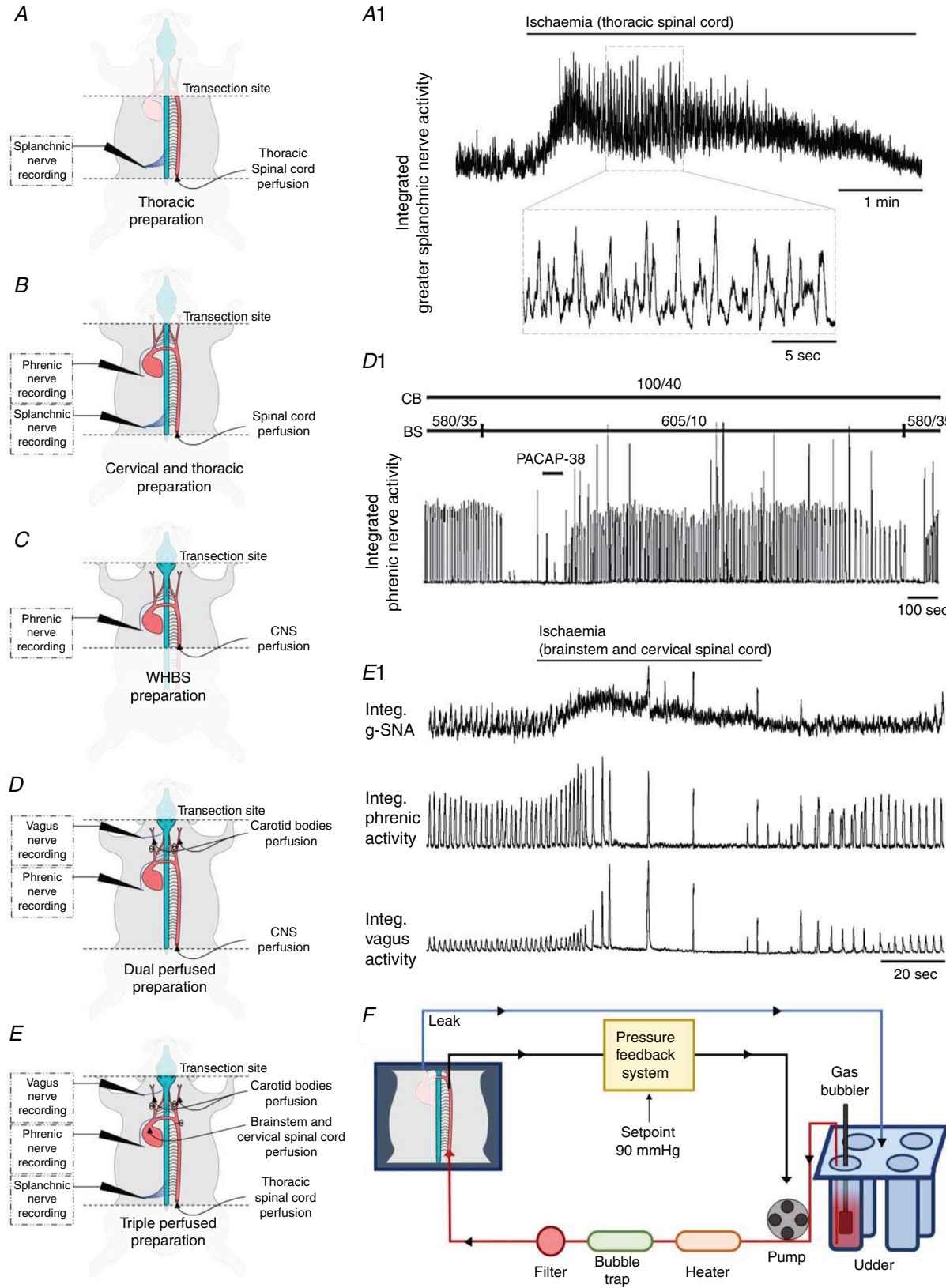

**Figure 12. The WHBP family**

WHBP descendants assist studies of integrated cardiorespiratory control. *A* and *A1*, thoracic preparation with data illustrating rhythmic splanchnic (sympathetic) bursts produced in response to specific thoracic spinal cord

ischaemia. *B*, cervical and thoracic preparation and *C*, WHBP, provided as a reference. *D* and *D1*, dual perfused preparation with data showing PACAP-38 activation of normoxic carotid bodies reverses central apnoea caused by specific brainstem hypocapnia (data from Fiamma et al., 2013). *E* and *E1*, triple perfused preparation with data showing greater splanchnic, phrenic and vagus nerve responses to specific brainstem ischaemia (carotid body and thoracic spinal cord maintained under baseline conditions). Note the hyperpnoea followed by apnoea and gasps concurrent with sympathetic activation. *F*, illustration of the 'udder' (exit of only one perfusion line is illustrated for clarity; red and blue are oxygenated and discharged perfusate, respectively) and PID pressure feedback system. The pressure feedback system is used initially to tune the preparation with a temperature-dependent pressure ramp and then to maintain the perfusion to the CNS compartment(s) at 90 mmHg. Modifications to the classic WHBP are: (1) the vagi and cardiac depressor nerves are ligated in the neck region; (2) sucrose replaces Ficoll as an oncotic agent as it is less expensive; (3) brainstem temperature is increased from 31°C to 33°C, and (4) brainstem $P_{CO_2}$ is increased from 35 to 40 Torr and perfusate $[K^+]$ is reduced from 5 to 4 mM.

can generate the majority of the systemic response produced by both chemoreflexes acting simultaneously; Day & Wilson, 2005, 2007; Fig. 12). This hypoadditivity permits the carotid body to restore phrenic activity when the brainstem is hypocapnic enough to elicit apnoea (<25 Torr; Day & Wilson, 2009; Fiamma et al., 2013). While this hypoadditivity has been observed consistently in subsequent rodent studies (e.g. Cummings, 2014; Tin et al., 2012), the interaction is hyperadditve in dogs *in vivo* (Blain et al., 2010); the reason for this species difference is not clear, but has been recently debated (Duffin & Mateika, 2013; Teppema & Smith, 2013; Wilson & Day, 2013; Wilson & Teppema, 2016). The DPP also proved pivotal in the discovery that the inflammatory signalling molecule lysophosphatidic acid activates the carotid bodies via TRPV1, triggering parasympathetically mediated bronchoconstriction; this discovery is of potential clinical importance suggesting a new treatment for asthma (Jendzjowsky et al., 2018).

Recently, the Wilson lab developed a triple perfused preparation (TPP) that involves three separate perfusion lines, so that the thoracic spinal cord, perfused via the descending aorta, is separate from the brainstem and carotid bodies (Fig. 12). Pressures to brainstem and spinal cord perfusions are regulated by separate PID feedback controllers. An example of the sympathetic response in this preparation to brainstem-specific ischaemia is illustrated in Fig. 12. Isolation of various segments of the spinal cord is also possible (Fig. 12). Employing these preparations has confirmed that specific spinal hypoxia and ischaemia causes reflex elevations in splanchnic and phrenic nerve activity, including the initiation of gasps, and has led to the first mechanistic understanding of spinal oxygen sensing (Barioni et al., 2022; Braga, Paton et al., 2007).

## Translating WHBP studies

Translation of work from the WHBP to *in vivo* rat studies as well as humans adds substantial credence to the WHBP, not only supporting the view that this is a physiologically relevant preparation but also demonstrating its power for

revealing putative mechanisms of disease. The following are a number of examples:

- Simms et al. (2009) showed a specific respiratory–sympathetic phase relationship that was distinct between the Wistar and SHR in the WHBP; this subtle phenotype is the same in the *in vivo* anaesthetised rats (Czyzyk-Krzeska & Trzebski, 1990).
- Zoccal et al. (2008) demonstrated that conditioning with chronic intermittent hypoxia led to forced expiratory activity (late expiratory abdominal discharge) that coupled to the sympathetic nervous system, which was not present in control animals; this was confirmed subsequently in awake rats (Moraes et al., 2013) and mainly during the wakefulness phase of the sleep–wake cycle (Bazilio et al., 2019).
- In the SHR, Traube–Hering waves are increased in the WHBP and contribute to arterial pressure (Simms et al., 2009); subsequently, these findings were confirmed in freely moving rats using radio-telemetry (Menuet et al., 2017).
- Menuet et al. (2020) demonstrated that optogenetic inhibition of pre-BötC neurons causes the same bradycardia and loss of RSA in the WHBP and *in vivo* under anaesthesia.
- Dutschmann et al. (2010), in Tau-P301L mice (a model of tauopathy), reported a shift of post-inspiratory laryngeal abduction into inspiration identified in the WHBP is associated with altered upper airway function *in vivo* in freely moving condition.
- The dependence of apnoeas on GABA deficiency and reduced serotonin 1a receptor stimulation was revealed in the WHBP of a murine Rett syndrome model; when these deficiencies were reversed, the apnoeas were substantially reduced *in vivo* (Abdala et al., 2010). Additionally, the change in central chemosensitivity in Rett mice was discovered in the WHBP and confirmed *in vivo* (Toward et al., 2013).
- Activation of the carotid body by lysophosphatidic acid (LPA) was shown to involve TRPV1 and LPA-specific receptors and induces bronchoconstriction in the WHBP and *in vivo* (Jendzjowsky et al., 2018). Attempts to translate this to humans are ongoing.

- Carotid body hyperexcitability of the SHR was characterised in the WHBP and confirmed *in vivo* (McBryde et al., 2013), as was the purinergic mechanism that underpins this hyperexcitability (Pijacka et al., 2016). Based on the predictions from the WHBP, hyperactive carotid bodies were demonstrated in humans with hypertension (Pijacka et al., 2016), which led to a first-in-human clinical trial (Narkiewicz et al., 2016).

## Possible future directions

The rhombomeric-like organisation of the medullary respiratory network has been established in the WHBP (Smith et al., 2007), but the question remains, does this extend into the pons (St-John & Bledsoe, 1985), such that pontine groups independently generate rhythmic activity? Pilot data support this contention (J. F. R. Paton & W. M. St-John). Future *in situ* studies might investigate if rhythmic trigeminal discharges, linked to that of the phrenic nerve, persist following separation of pons from medulla. This would constitute another neuronal oscillator and, if it exists, would lead to testing of its state-dependency and mechanism of operation.

The functional identification of respiratory neurons that directly modulate pre-sympathetic and/or parasympathetic cardiovascular neurons remains problematic, as any modulation of the activity of a respiratory neuron alters the whole respiratory network. Future technological developments, which will enable modulation of synaptic transmission alone, to alter only the connectivity between a targeted input respiratory neuron and a targeted output cardiovascular neuron, leaving the respiratory network unaltered, will address these issues. For example, a recently described pharmacogenetic tool enables inhibition of a targeted post-synaptic neuron by a targeted presynaptic neuron (Ngo et al., 2020). In addition, Gi/o-coupled photo-switchable G protein-coupled receptors are likely to provide powerful presynaptic inhibition (Copits et al., 2021; Mahn et al., 2021). These new developments should pave the way towards conclusive, functional identification of the central respiratory–cardiovascular connectome.

Although voltage-sensitive dyes have been utilised to image respiratory network transition from eupnoea to gasping (Potts & Paton, 2000), imaging approaches have been limited in the WHBP. The colourless blood free perfusate, negating light scattering, surely opens future opportunities. Mounting the WHBP under modern confocal or two-photon microscopes using transgenic animals with calcium-sensitive proteins, for example, would allow unprecedented new optogenetic and pharmacogenetic studies to unravel brainstem circuitry controlling visceral function in health and disease.

## Concluding remarks

Over the past 25 years, the WHBP has provided unparalleled opportunities to study the central mechanisms generating respiratory and autonomic activity, and respiratory–cardiovascular coupling. It continues to present opportunities to generate novel experimental methods to explore the functional neuro-anatomy of brainstem circuits regulating the rhythm of respiration, cardiovascular autonomic motor activity and sensory–motor integration of regulatory reflexes. Studies to date have collectively been able to mechanistically dissect the operation of these fundamental respiratory and cardiovascular control circuits. Many of them would not have been possible using conventional *in vivo* studies, as they take advantage of the unique access and control afforded by the WHBP. Importantly, many of these findings have been corroborated in subsequent *in vivo* animal and human studies, including first-in-human clinical trials supporting the benefit of the WHBP approach to undertake proof of concept studies of neural control circuits. As a reviewer of the original Paton (1996a) study remarked 'the *in situ* preparation is a technical *tour de force*'. Without doubt, it has enlightened new ways for our better understanding of the complexity of neural pathways in the brainstem. Finally, after reporting the preparation to The Physiological Society in Cork in 1995, a question from the audience asked how the preparation can be justified; we believe this review provides the answer.

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

## Additional information

### Competing interests

There are no competing interests, and all authors have no conflict of interest in accordance with journal policy.

### Author contributions

Each author has contributed written text to one or more sections of this review. All authors have edited the manuscript and contributed to revisions. All authors have read and approved the final version of this manuscript and agree to be accountable for all aspects of the work in ensuring that questions related to the accuracy or integrity of any part of the work are appropriately investigated and resolved. All persons designated as authors qualify for authorship, and all those who qualify for authorship are listed.

### Funding

A.P.A. – R01 AT008632 – CRCNS. A.M.A. – Australian Research Council (DP170104582) and Australian National Health and Medical Research Council (APP1156727). V.R.A. – São Paulo Research Foundation (FAPESP 2019/19894-8) and National Council for Development of Science and Technology (CNPq – Research Fellow: no. 304970/2017-4). R.R.D. – National Institutes of Health (U01 EB021960). J.D. – Clinical Fellowship from the Neurological Foundation of New Zealand. I.S.A.F. – University of Auckland PhD scholarship. B.H.M. – São Paulo Research Foundation (FAPESP-2018/15957-2) and the National Council for Development of Science and Technology (CNPq-309338/2020-4). C.M. – Agence Nationale de la Recherche (ANR-21-CE14-0009-01). D.J.A.M. – FAPESP (2019/11863-6 and 2021/06886-7) and CNPq (437375/2018-8 and 313719/2020-9). J.F.R.P. – Health Research Council of New Zealand (19/687) and the Sidney Taylor Trust. A.E.P. was funded by a Wellcome Trust Clinical research fellowship. J.C.S. – Intramural Research Program of the NIH, NINDS. T.A.D. – Natural Sciences and Engineering Research Council of Canada Discovery grants (NSERC RGPIN-2016-04915). R.J.A.W. – Canada Institute of Health Research (CIHR-201603PJT/366421). N.O.B. – SIDS Calgary Society.

### Acknowledgements

J.F.R.P. thanks the following for their support and encouragement during the development of the WHBP: Professor D. M. Armstrong (deceased); Dr Allan Levi (deceased), Professor Michael de Burgh Daly (deceased); Professor Max Headley; and Dr Julia Escardó-Paton. J.F.R.P. thanks the British Heart Foundation and Royal Society for originally funding the WHBP.

Open access publishing facilitated by The University of Auckland, as part of the Wiley – The University of Auckland agreement via the Council of Australian University Librarians.

### Keywords

cardiac ganglion, cardiac vagus, chemoreceptor, eupnoea, gasp, hypertension, hypoxia, Kölliker–Fuse, pre-Bötzinger complex, respiratory rhythm generation, sympathetic, sympathetic-respiratory coupling

### Supporting information

Additional supporting information can be found online in the Supporting Information section at the end of the HTML view of the article. Supporting information files available:

**Peer Review History**
**The WHBP set-up**

