## [Peer Review History · The Journal of Physiology]

Advancing respiratory-cardiovascular physiology with the working heart-brainstem preparation over 25 years

Julian F. R. Paton, Benedito H. Machado, Davi J. A. Moraes, Daniel B Zoccal, Ana Paula Abdala, Jeffrey C Smith, Vagner Roberto Antunes, David Murphy, Mathias Dutschmann, Rishi R Dhingra, Robin M McAllen, Anthony E Pickering, Richard J.A. Wilson, Trevor A Day, Nicole Orsi Barioni, Andrew M Allen, Clément Menuet, Joseph Donnelly, Igor S. A. Felipe, and Walter M St John

DOI: 10.1113/JP281953

Corresponding author(s): Julian Paton (j.paton@auckland.ac.nz)

The following individual(s) involved in review of this submission have agreed to reveal their identity: Rodrigo Del Rio (Referee #1); Ana C Takakura (Referee #2)

Review Timeline:

Submission Date:	20-Oct-2021
Editorial Decision:	25-Nov-2021
Revision Received:	15-Feb-2022
Accepted:	04-Mar-2022

Senior Editor: Ian Forsythe

Reviewing Editor: Frank Powell

Transaction Report:

Dear Julian,

Re: JP-TR-2021-281953 "Advancing respiratory-cardiovascular physiology with the working heart-brainstem preparation over 25 years" by Julian F. R. Paton, Benedito H. Machado, Davi J. A. Moraes, Daniel B Zoccal, Ana Paula Abdala, Jeffrey C Smith, Vagner Roberto Antunes, David Murphy, Mathias Dutschmann, Rishi R Dhingra, Robin M McAllen, Anthony E Pickering, Richard J.A. Wilson, Trevor A Day, Nicole Orsi Barioni, Andrew M Allen, Clément Menuet, Igor S. A. Felipe, and Walter M St John

Thank you for submitting your Topical Review to The Journal of Physiology. It has been assessed by a Reviewing Editor and by 2 expert referees and I pleased to tell you that it is considered to be acceptable for publication following satisfactory revision.

The reports are copied at the end of this email. Please address all of the points and incorporate all requested revisions, or explain in your Response to Referees why a change has not been made.

NEW POLICY: In order to improve the transparency of its peer review process The Journal of Physiology publishes online as supporting information the peer review history of all articles accepted for publication. Readers will have access to decision letters, including all Editors' comments and referee reports, for each version of the manuscript and any author responses to peer review comments. Referees can decide whether or not they wish to be named on the peer review history document.

I hope you will find the comments helpful and have no difficulty in revising your manuscript within 4 weeks.

Your revised manuscript should be submitted online using the links in Author Tasks Link Not Available. This link is to the Corresponding Author's own account, if this will cause any problems when submitting the revised version please contact us.

You should upload:

- A Word file of the complete text (including any Tables);
- An Abstract Figure, (with accompanying Legend in the article file)
- Each figure as a separate, high quality, file;
- A full Response to Referees;
- A copy of the manuscript with the changes highlighted.
- Author profile. A short biography (no more than 100 words for one author or 150 words in total for two authors) and a portrait photograph of the two leading authors on the paper. These should be uploaded, clearly labelled, with the manuscript submission. Any standard image format for the photograph is acceptable, but the resolution should be at least 300 dpi and preferably more.

- A 'Cover Art' file for consideration as the Issue's cover image;
- Appropriate Supporting Information (Video, audio or data set https://jp.msubmit.net/cgi-bin/main.plex?form_type=display_requirements#supp).

To create your 'Response to Referees' copy all the reports, including any comments from the Senior and Reviewing Editors into a Word, or similar, file and respond to each point in colour or CAPITALS. Upload this when you submit your revision.

I look forward to receiving your revised submission.

Yours sincerely,

Ian D. Forsythe
Deputy Editor-in-Chief
The Journal of Physiology
<https://jp.msubmit.net>
<http://jp.physoc.org>
The Physiological Society
Hodgkin Huxley House
30 Farringdon Lane
London, EC1R 3AW
UK
<http://www.physoc.org>
<http://journals.physoc.org>

EDITOR COMMENTS

Reviewing Editor:

This is a thorough review of an important experimental technique that has the potential to stimulate even more productive research. The referees both have comments and suggestions that you will want to consider to have the broadest influence and impact. However, the work is already very useful.

Senior Editor:

Congratulations on an interesting review, Julian and colleagues. In addition to the minor issues raised by the referees, can you re-write the abstract to contain more factual information about the topic of the review and avoid it reading like a proposal for the review, please.

See also 'Required Items' below.

REFeree COMMENTS

Referee #1:

In the present commissioned review, Paton and cols provided a very detailed view on the advances in physiology acquired by the used of the working heart-brainstem preparation (WHBP). Importantly, all the authors listed had used this preparation and the article benefit from their knowledge since is remarkably easy to follow. I would like to congratulate the authors by their work. I would also like to suggest some modifications to the text to try not to give a bias vision to the non-expert readership. I think that one important aspect that deserves a longer discussion is on whether all the findings (from the past and futures to come) collected through this preparation may recapitulate the physiology of whole animals. Authors extensively review several advancements in science where the prep has been instrumental. However, they also recognized some very important limitations which may or may not undermine the potential translation of the results obtained with this prep. For example, authors recognized that the WHBP did not work with older animals. Indeed, most of the studies that consider the WHBP has been done in young/juvenile animals. Then, how all the physiological data acquire with this prep could be applicable in older/bigger animals? This may also be relevant in pathological settings for example. Indeed, authors commented on hypertension and respiratory-sympathetic coupling. Hypertension develops in older people/animals and results obtained through the WHBP may not completely mimic the scenario in adult or older animals. Also as recognized by the authors, prep is perfused with low pressure and hyperoxic ringer, both conditions needed to keep the prep running but very far away from physiology. This also apply when comparing several other variables (i.e. blood shear stress, peripheral resistance, lower body sensory afferent activity, renal function, muscle metabo/mechanoreflexes). Then, I suggest to tempered authors conclusions over the whole manuscript and more clearly disclose the limitations of the findings using the prep in each specific section. Accordingly, a more specific limitation paragraph should be added in each section to clearly provide pros and cons of the WHBP, so reader can judge if their research may benefit or not by the use of the prep. The manuscript may benefit also if a more schematic figure showing pros and cons of the prep is added. All of these also apply for the section on "Possible future directions". Again, there are very nice ideas to develop but non of them has been put in context of whether they can or can't be translated into whole animal physiology.

Finally, for this reviewer not all the figures are actually needed. While I understand they provide more visual information they did not add anything new since most of them has been already published. Then, I would suggest to focus more on the different variations of the WHBP and the readouts that are possible to get with each one rather than illustrations of previous published material.

Referee #2:

The review by Dr. Julian Paton and colleagues reported that over the past 25 years, the WHBP (in situ preparation) has provided unparalleled opportunities to study the

central mechanisms related to the respiratory and autonomic activity. The preparation continues to present opportunities to generate novel experimental methods to explore the functional neuroanatomy of brainstem circuits regulating the rhythm of respiration, cardiovascular autonomic motor activity, and sensory-motor integration of regulatory reflexes. I am very impressed how the preparation impacts several scientists all over the Globe and how the preparation was important to elucidate the physiological mechanism. I have very few constructive comments to improve the review.

1) If the WHBP is so valuable to understand integrative physiology, why nowadays, we still have an aversive pool of scientist that still avoid using it? In addition, what are your thoughts about the fact that only 25 laboratories around the world use the preparation? Is not a small number considering the advantages of the preparation?

2) Low temperature may produce an unphysiological characteristic. What are the implications in terms of integrative physiology? For example, we know that many neuromodulators are released in the condition of low temperature. How is the influence of the neuromodulators on respiratory and cardiovascular parameters?

3) The WHBP was created mainly to study respiratory and cardiovascular regulation. Many manuscripts from the authors of the present study used the WHBP to understand cardiorespiratory coupling. Considering that the preparation is pretty much stable and worked at a low temperature and in a hyperoxic condition, how do both variables impact the physiological condition? In the same concept of cardiorespiratory integration, does the coupling change over time?

4) According to previous work in the literature (PMID: 23536061), it seems that postsynaptic inhibition within the preBötC and BötC is not essential for the generation of normal respiratory rhythm in intact mammals. The primary role of inhibition is in shaping the pattern of respiratory motor output, assuring its stability, and mediating reflex or volitional apnea. Is the difference observed in the WHBP (PMID: 27200412) mediated by a reduced preparation which impacts the inhibition on the generation of respiratory rhythm?

5) Although the WHBP is an important tool for the study of cardiorespiratory variables, it is important to emphasize that sleep-wake states have an impact on cardiovascular and respiratory control, especially on respiratory activity. At WHBP this does not seem to be possible and is difficult to assess. What would your comments be?

REQUIRED ITEMS:

-Please include an Abstract Figure. The Abstract Figure is a piece of artwork designed to give readers an immediate understanding of the Review Article and should summarise the main conclusions. If possible, the image should be easily 'readable' from left to right or top to bottom. It should show the physiological relevance of the Review so readers can assess the importance and content of the article. Abstract Figures should not merely recapitulate other figures in the Review. Please try to keep the diagram as simple as possible and without superfluous information that may distract from the main conclusion of the Review. Abstract Figures must be provided by authors no later than the revised manuscript stage and should be uploaded as a separate file during online submission labelled as File Type 'Abstract Figure'. Please ensure that you include the figure legend in the main article file. All Abstract Figures will be sent to a professional illustrator for redrawing and you may be asked to approve the redrawn figure before your paper is accepted.

-Your MS must include a complete "Additional information section" with the following 4 headings and content:

Competing Interests: A statement regarding competing interests. If there are no competing interests, a statement to this effect must be included. All authors should disclose any conflict of interest in accordance with journal policy.

Author contributions: Each author should take responsibility for a particular section of the study and have contributed to writing the paper. Acquisition of funding, administrative support or the collection of data alone does not justify authorship; these contributions to the study should be listed in the Acknowledgements. Additional information such as 'X and Y have contributed equally to this work' may be added as a footnote on the title page.

It must be stated that all authors approved the final version of the manuscript and that all persons designated as authors qualify for authorship, and all those who qualify for authorship are listed.

Funding: Authors must indicate all sources of funding, including grant numbers. If authors have not received funding, this must be stated.

It is the responsibility of authors funded by RCUK to adhere to their policy regarding funding sources and underlying research material. The policy requires funding information to be included within the acknowledgement section of a paper. Guidance on how to acknowledge funding information is provided by the Research Information Network. The policy also requires all research papers, if applicable, to include a statement on how any underlying research materials, such as data, samples or models, can be accessed. However, the policy does not require that the data must be made open. If there are considered to be good or compelling reasons to protect access to the data, for example commercial confidentiality or legitimate sensitivities around data derived from potentially identifiable human participants, these should be included in the statement.

Acknowledgements: Acknowledgements should be the minimum consistent with courtesy. The wording of acknowledgements of scientific assistance or advice must have been seen and approved by the persons concerned. This section should not include details of funding.

-Please upload separate high quality figure files via the submission form.

-Author profile(s) must be uploaded via the submission form. Authors should submit a short biography (no more than 100 words for one author or 150 words in total for two authors) and a portrait photograph of the two leading authors on the paper. These should be uploaded, clearly labelled, with the manuscript submission. Any standard image format for the photograph is acceptable, but the resolution should be at least 300 dpi and preferably more. A group photograph of all authors is also acceptable, providing the biography for the whole group does not exceed 150 words.

-It is the authors' responsibility to obtain any necessary permissions to reproduce previously published material
https://jp.msubmit.net/cgi-bin/main.plex?form_type=display_requirements#use

END OF COMMENTS

Confidential Review

20-Oct-2021

Response to Editor and Reviewers comments for:

JP-TR-2021-281953

"Advancing respiratory-cardiovascular physiology with the working heart-brainstem preparation over 25 years" by Julian F. R. Paton, Benedito H. Machado, Davi J. A. Moraes, Daniel B Zoccal, Ana Paula Abdala, Jeffrey C Smith, Vagner Roberto Antunes, David Murphy, Mathias Dutschmann, Rishi R Dhingra, Robin M McAllen, Anthony E Pickering, Richard J.A. Wilson, Trevor A Day, Nicole Orsi Barioni, Andrew M Allen, Clément Menuet, Igor S. A. Felipe, and Walter M St John

EDITOR COMMENTS

Please note: we have added Dr Joseph Donnelly to the authorship. Dr Donnelly is an ICU physician, and his contributions were delayed due to clinical duties related to COVID, staff shortages. The revised manuscript includes his contribution (p 7-8) and three additional references.

Reviewing Editor:

This is a thorough review of an important experimental technique that has the potential to stimulate even more productive research. The referees both have comments and suggestions that you will want to consider to have the broadest influence and impact. However, the work is already very useful.

Response:

We have responded fully to the reviewer's comments. In addition, to further broaden the reach and impact we have included an Appendix that contains engineering drawings and photographs of the equipment needed to set up the working heart brainstem preparation.

Senior Editor:

Congratulations on an interesting review, Julian and colleagues. In addition to the minor issues raised by the referees, can you re-write the abstract to contain more factual information about the topic of the review and avoid it reading like a proposal for the review, please.

See also 'Required Items' below.

Response:

Many thanks. We are all delighted with how well the review has been received. We have re-written the Abstract as suggested. We have responded fully to the reviewer's comments. In addition, to further broaden the reach and impact we have included an Appendix that contains engineering drawings and photographs of the equipment needed to set up the working heart brainstem preparation.

REFEREE COMMENTS

Referee #1:

In the present commissioned review, Paton and cols provided a very detailed view on the advances in physiology acquired by the used of the working heart-brainstem preparation (WHBP). Importantly, all the authors listed had used this preparation and the article benefit from their knowledge since is remarkably easy to follow. I would like to congratulate the authors by their work. I would also like to suggest some modifications to the text to try not to give a bias vision to the non-expert readership. I think that one important aspect that deserves a longer discussion is on whether all the findings (from the past and futures to come) collected through this preparation may recapitulate the physiology of whole animals. Authors extensively review several advancements in science where the prep has been instrumental. However, they also recognized some very important limitations which may or may not undermine the potential translation of the results obtained with this prep. For example, authors recognized that the WHBP did not work with older animals. Indeed, most of the studies that consider the WHBP has been done in young/juvenile animals. Then, how all the physiological data acquire with this prep could be applicable in older/bigger animals? This may also be relevant in pathological settings for example. Indeed, authors commented on hypertension and respiratory-sympathetic coupling. Hypertension develops in older people/animals and results obtained through the WHBP may not completely mimic the scenario in adult or older animals. Also as recognized by the authors, prep is perfused with low pressure and hyperoxic ringer, both conditions needed to keep the prep running but very far away from physiology. This also apply when comparing several other variables (i.e. blood shear stress, peripheral resistance, lower body sensory afferent activity, renal function, muscle metabo/mechanoreflexes). Then, I suggest to tempered authors conclusions over the whole manuscript and more clearly disclose the limitations of the findings using the prep in each specific section. Accordingly, a more specific limitation paragraph should be added in each section to clearly provide pros and cons of the WHBP, so reader can judge if their research may benefit or not by the use of the prep. The manuscript may benefit also if a more schematic figure showing pros and cons of the prep is added. All of these also apply for the section on "Possible future directions". Again, there are very nice ideas to develop but non of them has been put in context of whether they can or can't be translated into whole animal physiology.

RESPONSE:

Thank you for your thoughtful comments. To reduce our bias, we have: expanded the original 'Some limitations of the WHBP' paragraph and have added context-specific limitations paragraphs after two of the main sections (pages 11 & 15). We have also added a paragraph giving eight examples where data originally generated from the WHBP has translated to *in vivo* whole animal physiology, and examples where this has led to first-in-human clinical trials (p19). Incidentally, we know of no studies where data from the WHBP has not translated to *in vivo* conscious animals.

Finally, for this reviewer not all the figures are actually needed. While I understand they provide more visual information they did not add anything new since most of them has been already published. Then, I would suggest to focus more on the different variations of the WHBP and the

readouts that are possible to get with each one rather than illustrations of previous published material.

RESPONSE:

Thank you for the suggestion. We are all of the opinion that the Figures are valuable as they represent distinct data that can be obtained from WHBP variations. Six of the figures indicate either schematically and/or with a photograph a diverse number of preparation variants (Figs 1, 3, 6, 7, 9, 12). The other six Figures illustrate variants in terms of recording methods, different physiological systems that can be studied and at multiple levels of analysis. We have given these much thought so that representative examples of these different aspects of the preparation are illustrated.

PLEASE NOTE: We have added an Appendix giving technical drawings of the apparatus needed to set up the WHBP which will allow researchers to copy the customised parts unique to the extra corporal perfusion circuit and chamber of the WHBP. We believe this is a valuable addition to the Ms.

Referee #2:

The review by Dr. Julian Paton and colleagues reported that over the past 25 years, the WHBP (in situ preparation) has provided unparalleled opportunities to study the central mechanisms related to the respiratory and autonomic activity. The preparation continues to present opportunities to generate novel experimental methods to explore the functional neuroanatomy of brainstem circuits regulating the rhythm of respiration, cardiovascular autonomic motor activity, and sensory-motor integration of regulatory reflexes. I am very impressed how the preparation impacts several scientists all over the Globe and how the preparation was important to elucidate the physiological mechanism. I have very few constructive comments to improve the review.

Thank you for your careful evaluation of our review and the thoughtful comments you have suggested.

1) If the WHBP is so valuable to understand integrative physiology, why nowadays, we still have an aversive pool of scientist that still avoid using it? In addition, what are your thoughts about the fact that only 25 laboratories around the world use the preparation? Is not a small number considering the advantages of the preparation?

This is an interesting point. The twenty-five laboratories we quote are only the ones we are aware of; there may be others. The preparation requires some specialized equipment and expertise in small animal surgery; the latter may take some time to develop. Given this, we have added an Appendix that provides highly detailed drawings (and photographs) of the specialized equipment to facilitate easier adoption of the preparation. In the revised manuscript, and in response to Reviewer 1, we acknowledge the limitations of the WHBP; these may be sufficient to deter investigators. Notwithstanding, a driver for this review was to demonstrate the versatility and utility of the preparation and hope that this will encourage other investigators to use the WHBP. To encourage this, we have included an Appendix describing all the details required to make the custom equipment for the WHBP and have added a new section listing examples of how findings in the WHBP have predicted correctly numerous mechanisms operating both *in vivo* conscious

animals and humans, and have led to clinical trials. Perhaps this will temper the critics. In addition, we must take into account that we really don't know about the precise number of leading laboratories working on the central neural control of autonomic and respiratory functions around the world. Considering the citations in the references of most of the publications related to these fields, we believe that 25 laboratories using the WHBP is a close approximation to the number.

2) Low temperature may produce an unphysiological characteristic. What are the implications in terms of integrative physiology? For example, we know that many neuromodulators are released in the condition of low temperature. How is the influence of the neuromodulators on respiratory and cardiovascular parameters?

Originally, we chose 31°C as the temperature to run the WHBP but it will operate at body temperature. The lower temperature is preferred as it reduces metabolic rate and hence reduces oxygen demand, which is advantageous as an oxygen carrier in the perfusate is not needed. Clearly physiological processes such as synaptic release, action potentials, muscle contraction, heart and respiratory rates will be slower but not necessarily unphysiological. Slowing some processes down has assisted in their understanding and analysis. We have emphasised the lower temperature as a potential dis-advantage of the preparation (see p 6, 'Some limitations of the WHBP').

3) The WHBP was created mainly to study respiratory and cardiovascular regulation. Many manuscripts from the authors of the present study used the WHBP to understand cardiorespiratory coupling. Considering that the preparation is pretty much stable and worked at a low temperature and in a hyperoxic condition, how do both variables impact the physiological condition? In the same concept of cardiorespiratory integration, does the coupling change over time?

We understand the reviewer's point that the coupling of respiratory with cardiovascular autonomic activity may be affected by the "unphysiological condition" of the WHBP. This we have addressed by listing several examples (eight; see page 19) that show how data generated by the WHBP were translated successfully to *in vivo* animals (anaesthetised or awake) to reveal physiological and/or disease mechanisms. Moreover, we have given examples of how data from the WHBP predicted disease mechanisms in humans and led to first in human trials. From our experience to date, studies performed using the WHBP have all translated positively to *in vivo* animals. Finally, regarding a change in coupling with time: coupling does not change with time but does with disease.

4) According to previous work in the literature (PMID: 23536061), it seems that postsynaptic inhibition within the preBötC and BötC is not essential for the generation of normal respiratory rhythm in intact mammals. The primary role of inhibition is in shaping the pattern of respiratory motor output, assuring its stability, and mediating reflex or volitional apnea. Is the difference observed in the WHBP (PMID: 27200412) mediated by a reduced preparation which impacts the inhibition on the generation of respiratory rhythm?

This is a controversial point and one we wish to avoid as it is beyond the scope of this review. Some would argue that rhythm and pattern are distinct mechanisms whereas others would insist that pattern and rhythm are inseparable such that different patterns are formed by different

mechanisms driving the rhythm. The reference you quote (PMID 23536061) is an example disputing synaptic inhibition but there are many studies supporting it as essential; some examples are listed below:

- **Dhingra et al. (2019). Increasing local excitability of brainstem respiratory nuclei reveals a distributed network underlying respiratory motor pattern formation. *Front Physiol.* 23, 10:887.**
- **Marchenko et al. (2016). Perturbations of Respiratory Rhythm and Pattern by Disrupting Synaptic Inhibition within Pre-Botzinger and Botzinger Complexes. *eNeuro.* 3 e0011-16.**
- **Smith et al. (2007). Spatial and functional architecture of the mammalian brain stem respiratory network: a hierarchy of three oscillatory mechanisms. *J Neurophysiol.* 98, 3370-87.**

5) Although the WHBP is an important tool for the study of cardiorespiratory variables, it is important to emphasize that sleep-wake states have an impact on cardiovascular and respiratory control, especially on respiratory activity. At WHBP this does not seem to be possible and is difficult to assess. What would your comments be?

The preparation is decerebrate and typically this is at the pre-collicular level so studying the diurnal rhythm is not possible. However, it is possible to induce REM like sleep using carbachol injections into pontine regions as demonstrated previously Brandes et al. (2011; *Exp Physiol.* 96, 548-55). We have added this information to the revised manuscript on page 6.

REQUIRED ITEMS:

-Please include an Abstract Figure. The Abstract Figure is a piece of artwork designed to give readers an immediate understanding of the Review Article and should summarise the main conclusions. If possible, the image should be easily 'readable' from left to right or top to bottom. It should show the physiological relevance of the Review so readers can assess the importance and content of the article. Abstract Figures should not merely recapitulate other figures in the Review. Please try to keep the diagram as simple as possible and without superfluous information that may distract from the main conclusion of the Review. Abstract Figures must be provided by authors no later than the revised manuscript stage and should be uploaded as a separate file during online submission labelled as File Type 'Abstract Figure'. Please ensure that you include the figure legend in the main article file. All Abstract Figures will be sent to a professional illustrator for redrawing and you may be asked to approve the redrawn figure before your paper is accepted.

RESPONSE: Abstract Figure has been submitted

-Your MS must include a complete "Additional information section" with the following 4 headings and content:

Competing Interests: A statement regarding competing interests. If there are no competing interests, a statement to this effect must be included. All authors should disclose any conflict of interest in accordance with journal policy.

Author contributions: Each author should take responsibility for a particular section of the study and have contributed to writing the paper. Acquisition of funding, administrative support or the collection of data alone does not justify authorship; these contributions to the study should be listed in the Acknowledgements. Additional information such as 'X and Y have contributed equally to this work' may be added as a footnote on the title page.

It must be stated that all authors approved the final version of the manuscript and that all persons designated as authors qualify for authorship, and all those who qualify for authorship are listed.

Funding: Authors must indicate all sources of funding, including grant numbers. If authors have not received funding, this must be stated.

RESPONSE: This new section with the four heading has been added at the end o the manuscript.

It is the responsibility of authors funded by RCUK to adhere to their policy regarding funding sources and underlying research material. The policy requires funding information to be included within the acknowledgement section of a paper. Guidance on how to acknowledge funding information is provided by the Research Information Network. The policy also requires all research papers, if applicable, to include a statement on how any underlying research materials, such as data, samples or models, can be accessed. However, the policy does not require that the data must be made open. If there are considered to be good or compelling reasons to protect access to the data, for example commercial confidentiality or legitimate sensitivities around data derived from potentially identifiable human participants, these should be included in the statement.

Acknowledgements: Acknowledgements should be the minimum consistent with courtesy. The wording of acknowledgements of scientific assistance or advice must have been seen and approved by the persons concerned. This section should not include details of funding.

RESPONSE: Done

-Please upload separate high quality figure files via the submission form.

RESPONSE: Done

-Author profile(s) must be uploaded via the submission form. Authors should submit a short biography (no more than 100 words for one author or 150 words in total for two authors) and a portrait photograph of the two leading authors on the paper. These should be uploaded, clearly labelled, with the manuscript submission. Any standard image format for the photograph is acceptable, but the resolution should be at least 300 dpi and preferably more. A group photograph of all authors is also acceptable, providing the biography for the whole group does not exceed 150 words.

RESPONSE:

We request 3 co-authors:

Professor Benedito Machado – it was his idea to write the silver jubilee review of the WHBP.

Mr Igor Felipe – he is the next generation of scientists to use and further exploit the future benefits of the WHBP.

Professor Julian Paton – inventor of the WHBP.

-It is the authors' responsibility to obtain any necessary permissions to reproduce previously published material https://jp.msubmit.net/cgi-bin/main.plex?form_type=display_requirements#use

RESPONSES: We are seeking permission.

END OF COMMENTS

The Physiological Society is a company limited by guarantee. Registered in England and Wales, No. 00323575. Registered Office: Hodgkin Huxley House, 30 Farringdon Lane, London, EC1R 3AW, UK. Registered Charity No. 211585.

The Physiological Society and The Journal of Physiology are registered trademarks.

This email and any files transmitted with it are confidential and intended solely for the use of the individual or entity to whom they are addressed. If you have received this email in error please notify the sender. If you are not the named addressee you should not disseminate, distribute or copy this e-mail. The Physiological Society may monitor email traffic data.

The Physiological Society has taken reasonable precautions to ensure no viruses are present in this email, however does not accept responsibility for any loss or damage arising from the use of this email or attachments.

Dear Julian,

Re: JP-TR-2022-281953R1 "Advancing respiratory-cardiovascular physiology with the working heart-brainstem preparation over 25 years" by Julian F. R. Paton, Benedito H. Machado, Davi J. A. Moraes, Daniel B Zoccal, Ana Paula Abdala, Jeffrey C Smith, Vagner Roberto Antunes, David Murphy, Mathias Dutschmann, Rishi R Dhingra, Robin M McAllen, Anthony E Pickering, Richard J.A. Wilson, Trevor A Day, Nicole Orsi Barioni, Andrew M Allen, Clément Menuet, Joseph Donnelly, Igor S. A. Felipe, and Walter M St John

I am pleased to tell you that your Topical Review article has been accepted for publication in The Journal of Physiology, subject to any modifications to the text that may be required by the Journal Office to conform to House rules.

NEW POLICY: In order to improve the transparency of its peer review process The Journal of Physiology publishes online as supporting information the peer review history of all articles accepted for publication. Readers will have access to decision letters, including all Editors' comments and referee reports, for each version of the manuscript and any author responses to peer review comments. Referees can decide whether or not they wish to be named on the peer review history document.

The last Word version of the paper submitted will be used by the Production Editors to prepare your proof. When this is ready you will receive an email containing a link to Wiley's Online Proofing System. The proof should be checked and corrected as quickly as possible.

All queries at proof stage should be sent to tjp@wiley.com

The accepted version of the manuscript will be published online, prior to copy editing in the Accepted Articles section.

Are you on Twitter? Once your paper is online, why not share your achievement with your followers. Please tag The Journal (@jphysiol) in any tweets and we will share your accepted paper with our 22,000+ followers!

Best wishes

Ian

Ian D. Forsythe
Deputy Editor-in-Chief
The Journal of Physiology
<https://jp.msubmit.net>
<http://jp.physoc.org>
The Physiological Society
Hodgkin Huxley House
30 Farringdon Lane
London, EC1R 3AW
UK
<http://www.physoc.org>
<http://journals.physoc.org>

* IMPORTANT NOTICE ABOUT OPEN ACCESS *

Information about Open Access policies can be found here <https://physoc.onlinelibrary.wiley.com/hub/access-policies>

To assist authors whose funding agencies mandate public access to published research findings sooner than 12 months after publication The Journal of Physiology allows authors to pay an open access (OA) fee to have their papers made freely available immediately on publication.

You will receive an email from Wiley with details on how to register or log-in to Wiley Authors Services where you will be able to place an OnlineOpen order.

You can check if your funder or institution has a Wiley Open Access Account here <https://authorservices.wiley.com/author-resources/Journal-Authors/licensing-and-open-access/open-access/author-compliance-tool.html>

Your article will be made Open Access upon publication, or as soon as payment is received.

If you wish to put your paper on an OA website such as PMC or UKPMC or your institutional repository within 12 months of publication you must pay the open access fee, which covers the cost of publication.

OnlineOpen articles are deposited in PubMed Central (PMC) and PMC mirror sites. Authors of OnlineOpen articles are permitted to post the final, published PDF of their article on a website, institutional repository, or other free public server,

immediately on publication.

Note to NIH-funded authors: The Journal of Physiology is published on PMC 12 months after publication, NIH-funded authors DO NOT NEED to pay to publish and DO NOT NEED to post their accepted papers on PMC.

EDITOR COMMENTS

Reviewing Editor:

Thank you for this revision and excellent review of a unique experimental physiological preparation. Your work clearly explains its value to the field so far and hopefully this stimulates more good work, as well as the development of other integrative preparations to investigate important physiological problems.

Senior Editor:

Congratulations on an interesting review - I expect it to be well cited!

REFEREE COMMENTS

Referee #1:

Authors have successfully addressed all my suggestions. I appreciate the appendix since it should help new investigators to set-up the prep if it fits their research.

Referee #2:

This review analyzes the WHBP since its creation. It describes advantages, disadvantages, parameters that the WHBP enabled the investigation and future perspectives for it. This is a very well written review that covers all these aspects in a complete way. There is no doubt that this preparation contributed in many aspects to the understanding of the cardiorespiratory system. So, I congratulate the first author for developing it and all the authors for the document that will be of great value to the scientific community. I have no further suggestions, since the authors addressed all my comments.

1st Confidential Review

15-Feb-2022